# Dysfunction of duplicated pair rice histone acetyltransferases causes segregation distortion and an interspecific reproductive barrier

Ben Liao[1,2,6], You-Huang Xiang[1,6], Yan Li[3,6], Kai-Yang Yang[1,4], Jun-Xiang Shan [1,5], Wang-Wei Ye[1,5], Nai-Qian Dong [1,5], Yi Kan[1,5], Yi-Bing Yang[1,4], Huai-Yu Zhao[1,4], Hong-Xiao Yu[1,4], Zi-Qi Lu[1,2], Yan Zhao[3], Qiang Zhao[3], Dongling Guo[3], Shuang-Qin Guo[1,4], Jie-Jie Lei[1,4], Xiao-Rui Mu[1,4], Ying-Jie Cao[1,4], Bin Han [3] ✉ & Hong-Xuan Lin [1,2,4,5] ✉

Postzygotic reproductive isolation, which results in the irreversible divergence of species, is commonly accompanied by hybrid sterility, necrosis/weakness, or lethality in the $F_1$ or other offspring generations. Here we show that the loss of function of *HWS1* and *HWS2*, a couple of duplicated paralogs, together confer complete interspecific incompatibility between Asian and African rice. Both of these non-Mendelian determinants encode the putative Esa1-associated factor 6 (EAF6) protein, which functions as a characteristic subunit of the histone H4 acetyltransferase complex regulating transcriptional activation via genome-wide histone modification. The proliferating tapetum and inappropriate polar nuclei arrangement cause defective pollen and seeds in $F_2$ hybrid offspring due to the recombinant HWS1/2-mediated misregulation of vitamin (biotin and thiamine) metabolism and lipid synthesis. Evolutionary analysis of *HWS1/2* suggests that this gene pair has undergone incomplete lineage sorting (ILS) and multiple gene duplication events during speciation. Our findings have not only uncovered a pair of speciation genes that control hybrid breakdown but also illustrate a passive mechanism that could be scaled up and used in the guidance and optimization of hybrid breeding applications for distant hybridization.

Reproductive isolation (RI) hinders gene flow between intraspecific or interspecific populations during speciation, therefore contributing to the maintenance of species identity[1,2]. Hybrid breakdown (HB), a typical form of RI, is defined as reduced hybrid viability and/or fertility segregation in the $F_2$ or the later generation. Differing from other hybrid incompatibilities, HB involves intrinsic postzygotic reproductive barriers, such as hybrid inviability (including weakness[3], necrosis[4], and chlorosis[5]) and hybrid sterility (in the male, female, or both

[1]National Key Laboratory of Plant Molecular Genetics, CAS Centre for Excellence in Molecular Plant Sciences, Shanghai Institute of Plant Physiology and Ecology, Chinese Academy of Sciences, Shanghai 200032, China. [2]School of Life Science and Technology, ShanghaiTech University, Shanghai 201210, China. [3]China National Center for Gene Research, National Key Laboratory of Plant Molecular Genetics, CAS Center of Excellence in Molecular Plant Sciences, Shanghai Institute of Plant Physiology and Ecology, Chinese Academy of Sciences, Shanghai 200233, China. [4]University of the Chinese Academy of Sciences, Beijing 100049, China. [5]Guangdong Laboratory for Lingnan Modern Agriculture, Guangzhou 510642, China. [6]These authors contributed equally: Ben Liao, You-Huang Xiang, Yan Li. ✉e-mail: bhan@ncgr.ac.cn; hxlin@cemps.ac.cn

gametes). The classical Bateson-Dobzhansky-Muller (BDM) model partially addresses hybrid sterility, the most common form of post-zygotic barrier that is attributed to deleterious genetic interactions between incompatible allelic variations, but fails to consider the divergent evolution process[6–8]. A widely accepted viewpoint that has been proposed to reasonably explain the biased allele transmission and disharmonious effect conferring plant hybrid sterility is the killer-protector system[9,10]. In broad terms, genomic conflicts involved in hybrid incompatibility that occurs in sporophytes and gametophytes are basically divided into two types; single or dual locus-controlled $F_1$/$F_2$ male/female sterility models[11–13]. However, only a limited number of barrier genes have been successfully cloned from rice at present[14–24], and the molecular and evolutionary mechanisms underlying the reproductive barrier and compatibility are unknown[25]. The triallelic *S1* system is a regulator that governs RI in hybrids produced by crossing Asian rice (*Oryza sativa* L.) and African rice (*O. glaberrima* Steud.) and fits the typical one-locus allelic interaction model[26–28]. Nevertheless, it has not yet been possible to elucidate the complicated genetic basis of RI in all scenarios of evolution and domestication for African rice. Also, large gaps remain in our understanding of underappreciated mechanisms for activity-dependent epigenetic regulation of RI in rice.

ILS, which commonly occurs on the condition of failed coalescence and is in the form of retained genetic polymorphisms when new lineages rapidly descend from the ancestors, has been an enigmatic source of topological discordance between gene trees and species trees[29]. This evolution event which can easily be ignored and confused with introgression because of similar patterns of shared genetic diversity, often disrupts our understanding of the relationship between genomic and phenotypic divergence during speciation. Notwithstanding, emerging evidence regarding to ILS has been reported both in plants[30,31] and in animals[32–34], the general impact of ILS has remained elusive, and little is reported about the influences of ILS in the evolution process of rice. In this study, we systematically explore the evolutionary trajectory and describe two transposon-mediated duplicated recessive lethal genes, *hws1^C^* from *japonica* rice and *glaberrima*-like *hws2^Null^* that cause anomalous transcription through inoperative epigenetic regulation, triggering complete sterility and weakness in the $F_2$ or later generations during allopatric speciation of the related *Oryza* species.

## Results

### Loss-of-function mutations in an interactive gene pair confer hybrid incompatibility

In constructing a rice chromosome segment substitution line (CSSL) library derived from an interspecific cross with *O.glaberrima* (CG14; genotype defined as G) genomic segments in the *O. sativa japonica* variety 'Wuyunjing7' (WYJ7; genotype defined as W) (Supplementary Fig. 1a–c), a CG14-originated allele (defined as *hws2^CG14^*) of the *Hybrid Weakness and Sterility 2* locus (abbreviated 'HWS2' hereafter) on chromosome 12 displayed eliminated transmission of gametes (Fig. 1a and Supplementary Table 1), and its positive carriers (genotype G2G2) segregating from *HWS2* heterozygotes (genotype W2G2) exhibited complete sterility along with a suite of weakness symptoms, including growth retardation and developmental disorders (Fig. 1b). These individuals inevitably produced small seedlings or ceased to grow afterward, and some even failed to germinate, indicating that *HWS2* controls post-embryonic developmental competence and seed vigor as well (Supplementary Fig. 1d). Examination of the self-pollinated progeny of *HWS2* hybrids showed that sterile plants with the homozygous CG14-type genotype (G2G2) were barely detected, whereas plant homozygous for *HWS2^WYJ7^* (genotype W2W2) or the W2G2 plants developed normally and grew vigorously with high seed-setting rates (Fig. 1a, b). The segregation ratio in the selfing progeny was 7:8:1 (99:113:14) (W2W2:W2G2:G2G2), and Chi-square independence tests (Fig. 1a and Supplementary Table 1) showed that the CG14-type

gametes at *HWS2* locus were not independently transmissible to the progeny and that segregation was not in accordance with Mendel's laws of inheritance. Intriguingly, the lines in this CSSL library carrying the *HWS2* locus that we eventually obtained all maintained normal spikelet fertility without the stunted phenotype. We therefore speculated that digenic epistatic interactions may exist, and contribute to abnormal gametophyte development and severe segregation distortion. We then made use of a whole genome survey of interacting loci using independence tests of marker segregations in an $F_2$ population from a cross between a selected CSSL line harboring the CG14-derived *hws2* segment and its recurrent parent (SG178 × WYJ7) (Supplementary Fig. 1a). Genetic analysis revealed that the hereditary sterility was conditioned by another interactive locus named *Hybrid Weakness and Sterility 1* (*HWS1*) on chromosome 1. *HWS1* acted in a similar manner in transmission ratio distortion, while the *HWS1^CG14^* allele was predominately selected for preferential transmission in contrast to the *HWS2* locus (Fig. 1a and Supplementary Table 1). The underlying genes (*HWS1* and *HWS2*) responsible for the inductive hybrid failure were subsequently delimited to 35.25-kb and 337.5-kb regions, respectively (Supplementary Fig. 1e, f).

Within the *HWS1* genomic interval, there were three annotated candidate genes, *LOC_Os01g13229*, *LOC_Os01g13250*, and *LOC_Os01g13260* (*229*, *250*, and *260* for short) (Supplementary Fig. 1e). The flanking genes, *229* and *260*, both encode cyclin-A1 proteins, which are thought to be expressed during mitosis and hence regulate the G2/M transition. Previous studies have shown that there are at least 49 predicted cell cyclin genes in rice[35], some of which were reported to be involved in seed development and thereby lead to empty compressed grains[36]. We thus generated *229/260* double-knockout mutants in disparate genetic backgrounds (Supplementary Fig. 2a). However, these loss-of-function mutants failed to perfectly mimic absolute infertility regardless of whether they were in the SG178 (*hws2* homozygote harboring the nonfunctional *hws2^CG14^*) or WYJ7 (*HWS2* homozygote containing the functional *HWS2^WYJ7^*) background, although the functional deficiency caused abnormal seeds and increased awn length in both genetic backgrounds (Supplementary Fig. 2b), which indicates that neither are the expected causal genes for the incompatibility syndrome. We next validated the identity of *HWS1* by editing the *250* gene likewise in both the SG178 and WYJ7 genetic backgrounds (Supplementary Fig. 2c). The resulting knockout lines in the SG178 background (*HWS1^CG14^*/*hws2^CG14^*), free of the Cas9 construct after segregation, manifested an identical highly sterile phenotype similar to NIL-*hws1^WYJ7^*/*hws2^CG14^* (Fig. 1c, g, h). Meanwhile, compared to the wild-type, the independent lines carrying defective *HWS1* alleles in the WYJ7 background (*hws1^WYJ7^*/*HWS2^WYJ7^*) exhibited no obviously visible phenotypes (Supplementary Fig. 2d). These results showed that the disrupted *250* gene is responsible for the *HWS1*-mediated reproductive barrier.

Since significant recombination repression occurs in the centromeric region, we can hardly shorten the candidate region of *HWS2* locus further simply by large-scale screening of recombinant individuals (Supplementary Fig. 1f). Comparative studies of the CG14 and WYJ7 genomes showed that the majority of predicted genes in this region encode retrotransposons or proteins without functional annotation except for only a very few candidate genes including *LOC_Os12g20310* (hereafter known as *310*) and the adjacent *LOC_Os12g20324* (Supplementary Fig. 2e). Both of the above shared quite a high homology with *250* and *260*, respectively. We thereby sequenced the genomes of CG14 and WYJ7 and found that *310* was entirely lost due to the absence of this duplicated chromosomal segment in CG14 (Fig. 1d and Supplementary Fig. 2f). The Dobzhansky-Muller-type incompatibility syndrome in plants is typically explained by the detrimental interference among at least two genetic loci with reciprocal replication[37]. Accordingly, we generated the null allele of *310* into the WYJ7 background

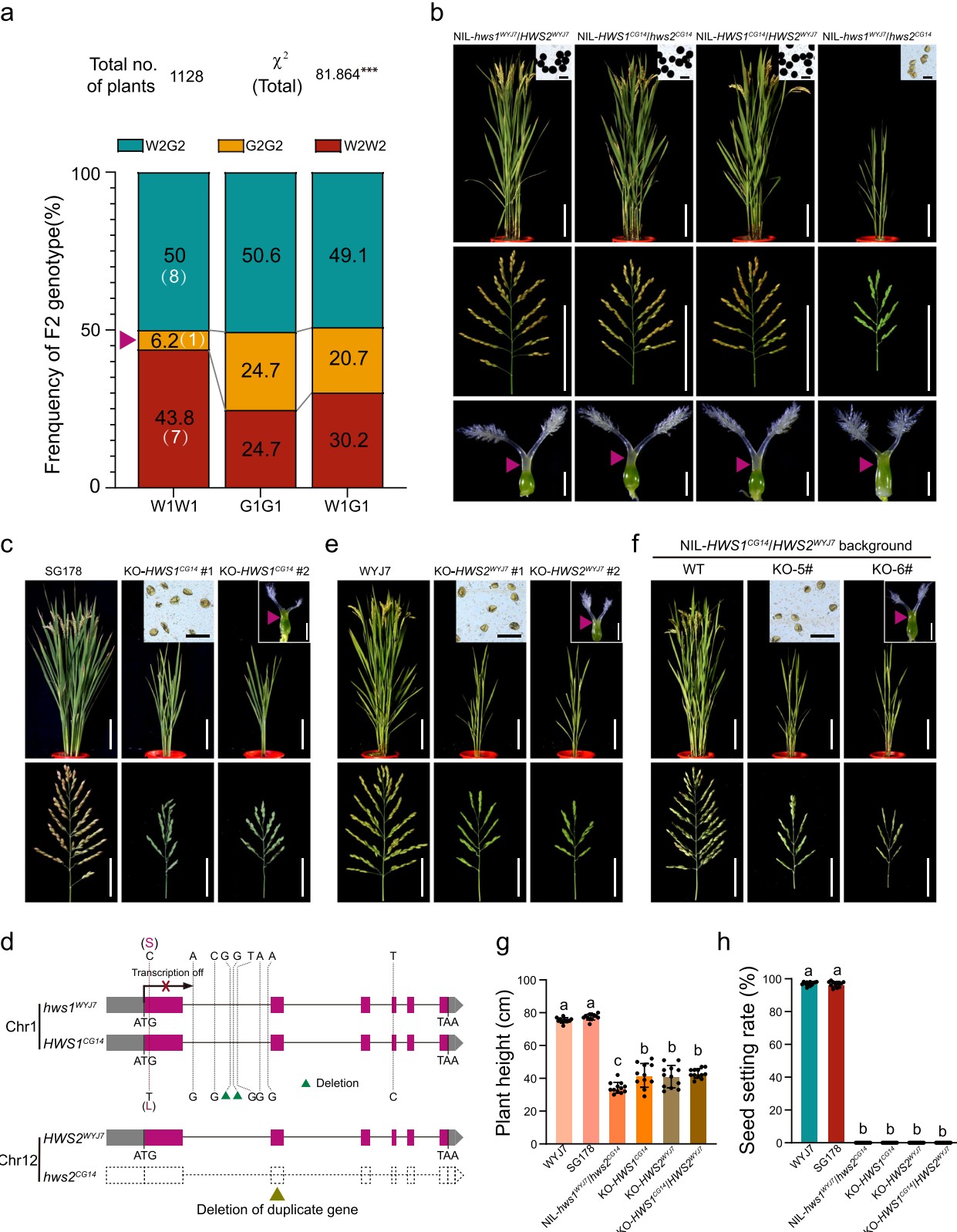

(*hws1^{WYJ7}/HWS2*) and evaluated its effects on reproduction (Supplementary Fig. 2c). As anticipated, the dysfunctional *310* allele successfully triggered pollen and spikelet abortion along with stunted growth (Fig. 1e, g, h), indicating that the paralogous *310* gene constitutes another component of the reproducible interaction. Furthermore, application of gene editing to simultaneously inactive both *250* and *310* effectively initiated aberrant gametophyte development in the fertile NIL-*HWS1^{CG14}/HWS2^{WYJ7}* homozygote background (Fig. 1f–h and Supplementary Fig. 2c), but inactivating either of the two normal copies, *HWS1^{CG14}* and *HWS2^{WYJ7}*, did not induce the sterility and weakness syndrome and the mutant plants showed full seed set full seeds (Supplementary Fig. 2c, g). These results indicate that a viable *glaberrima-japonica* hybrid requires at least one functional copy of either *HWS1* or *HWS2*.

**Fig. 1 | The combination of two silenced duplicated copies of *OsEAF6* contributes to reproductive barriers in rice. a** The resulting segregation ratios of $F_2$ genotypes at the barrier loci in self-pollinated progeny plants derived from a cross between WYJ7 and SG178. The *HWS2* genotypes are given by analogy with *HWS1*. \*\*\*$p = 2.06E\text{-}14 < 0.001$ in the two-sided $\chi^2$ test. **b** Mature plant architecture (upper panels), panicle morphology (middle panels), and ovules indicated with pink triangles after dissection (lower panels) in pairwise *HWS1/2* NILs. $I_2$-KI stained pollen grains indicating fertility are shown in the top right corners. Scale bars = 15 cm/10 cm/1 mm/100 µm in the upper/middle/bottom/top right corner panels, respectively. **c** Plants (top row, 10 cm), pollen grains (inset in the middle, 100 µm), floral organs (inset in the right, 1 mm), and spikelets (bottom row, 2.5 cm) of wild type (SG178) and *HWS1* knockout (KO) lines in the SG178 background at maturity. **d** Gene structure diagrams and sequence comparison of the respective *HWS1/2* alleles from WYJ7 and CG14. The dotted red line shows the position of a single-nucleotide

substitution mutation and the corresponding amino acid replacement produced by this base change in *hws1WYJ7* and *HWS1CG14*. The coding region and intron SNPs with non-synonymous substitutions are marked with black dotted lines. The green and dark yellow arrowheads indicate the base deletion and deleted segmental duplication of *HWS2* in CG14, respectively. Images of the wild-type control and transgenic plants [the single *hws2* (**e**) and *hws1/hws2* double mutants (**f**)]. Scale bars = 15 cm (top rows), 10 cm (bottom rows), and 1 mm/100 µm in respective insets (top right corner). Statistical comparisons of plant height (**g**) and seed setting rate (**h**) in the parental lines and the *HWS1/2* loss-of-function (KO) mutants. Data are means ± SD ($n = 12$ (WYJ7, SG178, NIL-*hws1WYJ7*/*hws2CG14*, KO-*HWS2WYJ7*, and KO-*HWS1CG14*/*HWS2WYJ7*) or 11 plants (KO-*HWS1CG14*)). Different letters denote significant differences ($p < 0.05$, one-way ANOVA with two-sided Tukey's HSD test). *P* values for **g**, **h** are adjusted and shown in the Source Data file.

The coupled *HWS* genes are both predicted to encode OsEAF6 which contains a highly conserved EAF6 domain in evolution (Supplementary Fig. 3a, b). Sequence alignments of the matched NIL-*HWS1* and NIL-*HWS2* genomes suggested that the *HWS1CG14* and *HWS2WYJ7* alleles had the same protein sequences except for two amino acid substitutions (Leu[20] to Ser, Asp[164] to Asn), both of which were far away from the functional EAF6 domain (Supplementary Fig. 3c). The first amino acid replacement near the N-terminus also occurred in the nucleotide sequence of *HWS1* between WYJ7 and CG14 (Fig. 1d). We examined the spatiotemporal expression patterns and levels of the diverged *HWS1/2* alleles and found that *HWS1CG14* and *HWS2WYJ7* were ubiquitously expressed in vegetative organs and anthers at various developmental stages in *HWS1* or *HWS2* fertile plants, whereas the transcripts of *hws1WYJ7* were almost undetectable in any tissues of their sterile counterparts (Supplementary Fig. 4a, b). The lack of *HWS1* transcripts caused by failure of expression in WYJ7 due to promoter mutation (Supplementary Fig. 4c, see below) and the absence of the duplicated segment containing *HWS2* in CG14 suggest that both *hws1WYJ7* and *hws2CG14* are loss-of-function alleles that render 100% stable sterility in rice. Additional RNA in-situ hybridization assays showed a similar trend that strong signals of *HWS1CG14* were detected in meiotic cells while that of *hws1WYJ7* were undetectable (Supplementary Fig. 5). Collectively, the appearance of sterile individuals in the progeny of a *glaberrima-japonica* cross is controlled by a pair of recessive genetic elements behaving segregation distortion.

## Morphological and cytological bases of male and female sterility mediated by the *HWS1/2* gene pair

We conducted reciprocal cross experiments to investigate the characteristics of hybrid breakdown in the sterile segregants. The sterile plants (NIL-*hws1WYJ7*/*hws2CG14*) were used in reciprocal crosses with WYJ7 and SG178, and no single seed was obtained from those crosses (Supplementary Fig. 6a–d). Instead, the control plants (*HWS1CG14*/*hws2CG14*) produced vigorous seeds after saturation pollination with SG178 (Supplementary Fig. 6e, f). These results suggest that the stable dysgenesis is attributed to both pollen and embryo-sac infertility. To decipher the basis of male sterility, we examined semi-thin sections of anthers throughout the process of pollen development (Fig. 2a–l) and observed, in NIL-*hws1WYJ7*/*hws2CG14* plants, a severely swollen and asymmetrical tapetum along with irregular microspores from stages 10 to 14 (Fig. 2i–l). Compared to fertile controls (WYJ7) possessing round and deeply stained pollens (Fig. 2f), the inward-growing tapetum tightly squeezed tetrads and made them distorted leading to defective pollens thereafter in NIL-*hws1WYJ7*/*hws2CG14* plants (Fig. 2l).

Ultrastructural studies of the tapetal layer showed that the tapetal membrane of WYJ7 was distinctly decomposed and precipitated around Ubisch bodies at stage 10 (Fig. 2m). A typical two-layer wall structure composed of the tectum, bactum, and nexine was also clearly visible (Figs. 2n, n1). In contrast, in the chamber of sterile anthers of NIL-*hws1WYJ7*/*hws2CG14* plants, there were no clear and intact

Ubisch bodies observed on the inner tangential wall of the tapetum (Fig. 2o). Remarkably large vacuoles in the thickened tapetum were irregularly formed during the process of tapetal disintegration (Fig. 2o). Moreover, parts of the exine in microspores showed abnormal protuberances with uneven bactum (Figs. 2p, p1). Consistent with the above observations, the main cytological reason behind pollen abortion is considered to be the stimulated proliferation of the tapetal cells, which interferes with microspore development. Scanning electron microscopy (SEM) observations further visualized the overall appearance features of anthers and pollen grains at maturity, revealing that abortion in *hws1WYJ7*/*hws2CG14*-type pollen grains is possibly due to their shrunken morphology and poorly-developed anther wall (Figs. 2q, q1-q8).

Dynamic examination of embryo sac development in NIL-*hws1WYJ7*/*hws2CG14* plants showed no obvious defects with respect to structure and morphology prior to the eight-nucleate embryo sac stage (Fig. 2r, s). However, the two polar nuclei occasionally failed to correctly locate and form a binucleate central cell near the micropylar end during later stages of development (Fig. 2t-w). These abortive and smaller embryo sacs were also found to be developmentally arrested, either empty or degenerated, in comparison with those in the fertile control (WYJ7) plants (Fig. 2v, w). Based on these results, we concluded that inappropriate polar nuclei arrangement, stagnant development, and abnormal cellularization could result in the inability to form mature embryo sacs during female gamete development in NIL-*hws1WYJ7*/*hws2CG14* plants.

## Dysfunctions in HWS1/2 contribute to H4Ac loss and transcriptomic reprogramming misregulation

EAF6 is a critical component of the nucleosomal acetyltransferase of the H3/H4 (NuA3/NuA4) complex and has been reported to participate in extensive transcriptional regulation via nuclear H4 or H2A acetylation, thereby affecting plant viability[38–40]. Subcellular localization analysis indicated that HWS1 and HWS2 are specifically targeted to the nuclei (Fig. 3a). The in vitro histone acetylation (HAT) assay showed that HWS2 is able to acetylate a histone H4 peptide (Fig. 3b). Fractionation of chromatin followed by western blotting using antibodies against various acetylation sites in the histone H2A, H3 and H4 N-terminal tail showed a pronounced decline in acetyl-H2A and acetyl-H4 in NIL-*hws1WYJ7*/*hws2CG14*, and this decline was most prominent at lysine 5 and 9 of histone H2A (H2AK5Ac and H2AK9Ac) and at lysine 5 of histone H4 (H4K5Ac) among all eight histone H2A and H4 lysine residues tested (Fig. 3c). To gain more insight into the correlation between the loss of H4Ac and transcriptional reprogramming, we first performed RNA deep sequencing (RNA-seq) and identified 19,806 differentially-expressed genes (DEGs) (fold-change >2) between the two corresponding NILs (Supplementary Fig. 7a). The enriched terms from combined Gene Ontology (GO) classification and Kyoto Encyclopedia of Genes and Genomes (KEGG) pathways analysis showed that the statistically significant DEGs were primarily associated with

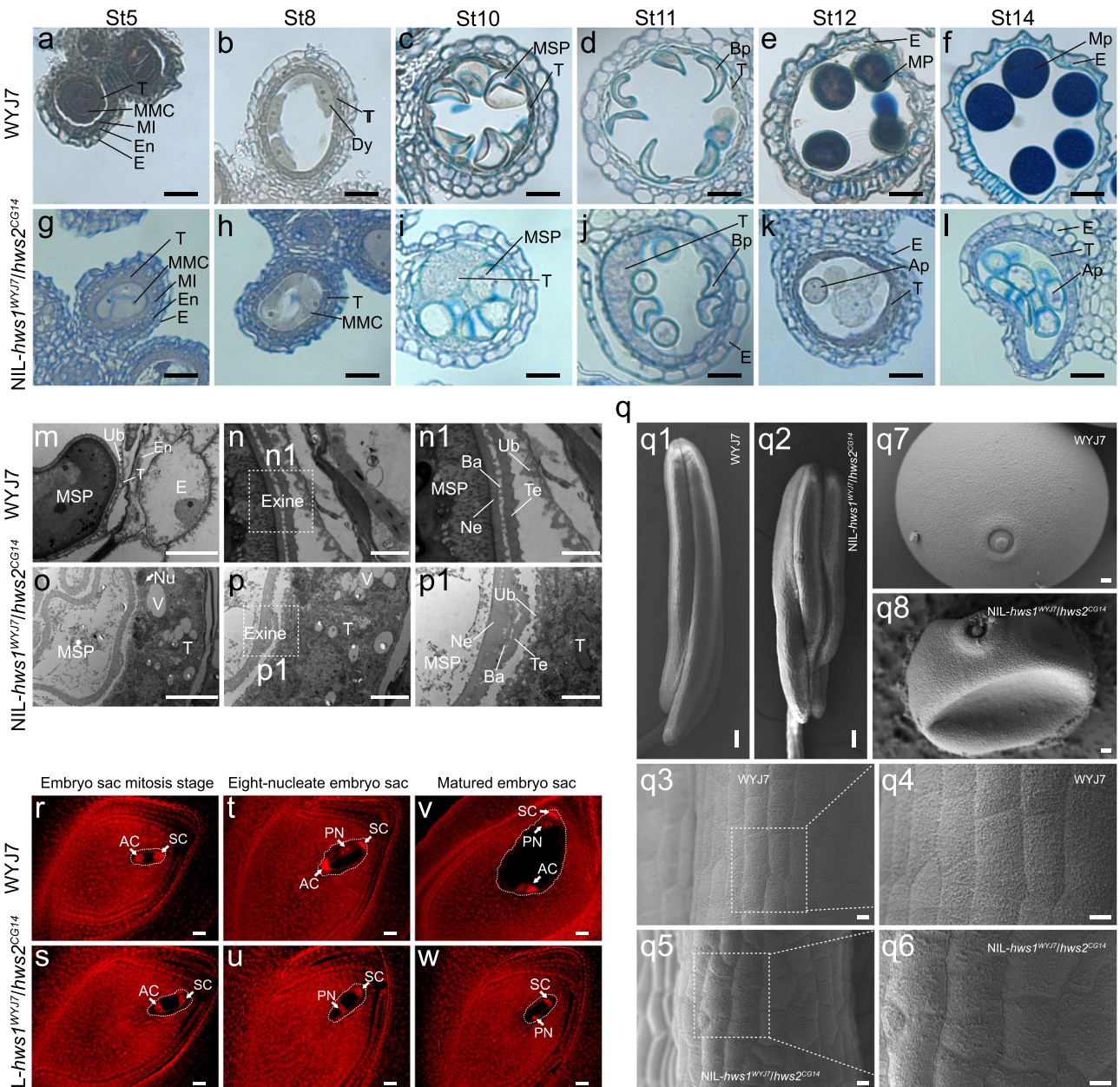

**Fig. 2 | Reciprocal gene loss of the duplicated HWS pair causes abortive pollen and lethal embryos in hybrid progeny.** Transverse sections of anthers at various developmental stages from WYJ7 (**a–f**) and NIL-*hws1*^WYJ7/*hws2*^CG14 (**g–l**). St5, secondary sporogenous cell stage; St8, dyad stage; St10, vacuolated microspore stage; St11, bicellular pollen stage; St12, tricellular pollen stage; St14, mature pollen stage. E, epidermis; En, endothecium; MI, middle layer; T, tapetum; MMC, meiocyte mother cell; Dy, dyad cell; Msp, microspore; Bp, bicellular pollen; Mp, mature pollen; Ap, abortive pollen. Scale bars = 20 μm. TEM observations showing anthers at St10 of WYJ7 (**m, n**) and NIL-*hws1*^WYJ7/*hws2*^CG14 (**o, p**). The photomicrographs with labels **n1, p1** are enlargements of the areas enclosed by white dotted lines in **n, p**. E, epidermis; En, endothecium; Msp, microspore; T, tapetum; V, vacuole; Nu, nucleus; Ne, nexine; Te, tectum; Ba, bactum; Ub, Ubisch bodies. Scale bars = 10 μm (**m, o**), 5 μm (**n, p**), and 2 μm (**n1, p1**). **q** SEM images of the normal-shaped anther (**q1, q3, q4**) and normal pollen grain (**q7**) from WYJ7, and shrunken anther (**q2, q5, q6**) and invaginated pollen grain (**q8**) from a NIL-*hws1*^WYJ7/*hws2*^CG14 plant. Epidermal cells and regions of higher magnification (boxed) are shown in **q3-q6**. Scale bars = 100 μm (**q1, q2**), 10 μm (**q3-q6**), and 2 μm (**q7, q8**). Microscopic examination of developing (**r-u**) and mature embryo sacs (**v-w**) of WYJ7 (**r, t, v**) and NIL-*hws1*^WYJ7/*hws2*^CG14 (**s, u, w**). PN, polar nucleus; AC, antipodal cell; SC, synergid cell. Scale bars = 100 μm. All experiments were repeated independently at least three times of at least three plants, with similar results.

mRNA surveillance, fatty acid, lipid metabolism, RNA transport, and small molecule binding process (Supplementary Fig. 7b, c). As shown by heat maps, the expression levels of several representative male determinants related to fatty acid synthesis and lipid transport were substantially down-regulated in NIL-*hws1*^WYJ7/*hws2*^CG14 plants (Supplementary Fig. 7d)[41–45], which may account for the adverse formation of the enlarged tapetal zone, unshaped pollen exine, and unusual cuticle surface.

To further visualize the in vivo differences in the degree of acetylated histone H4 at the whole-genome level, a chromatin immunoprecipitation sequencing (ChIP-seq) experiment was unitedly employed to analyze the transcriptome data. Strikingly, the average H4 occupancy in NIL-*hws1*^WYJ7/*hws2*^CG14 displayed a dramatic suppression within the peak downstream of the transcriptional start site (TSS) (Fig. 3d and Supplementary Fig. 7e). A total of 2,297 H4Ac enriched peaks corresponding to 2,175 genes were identified, the vast majority

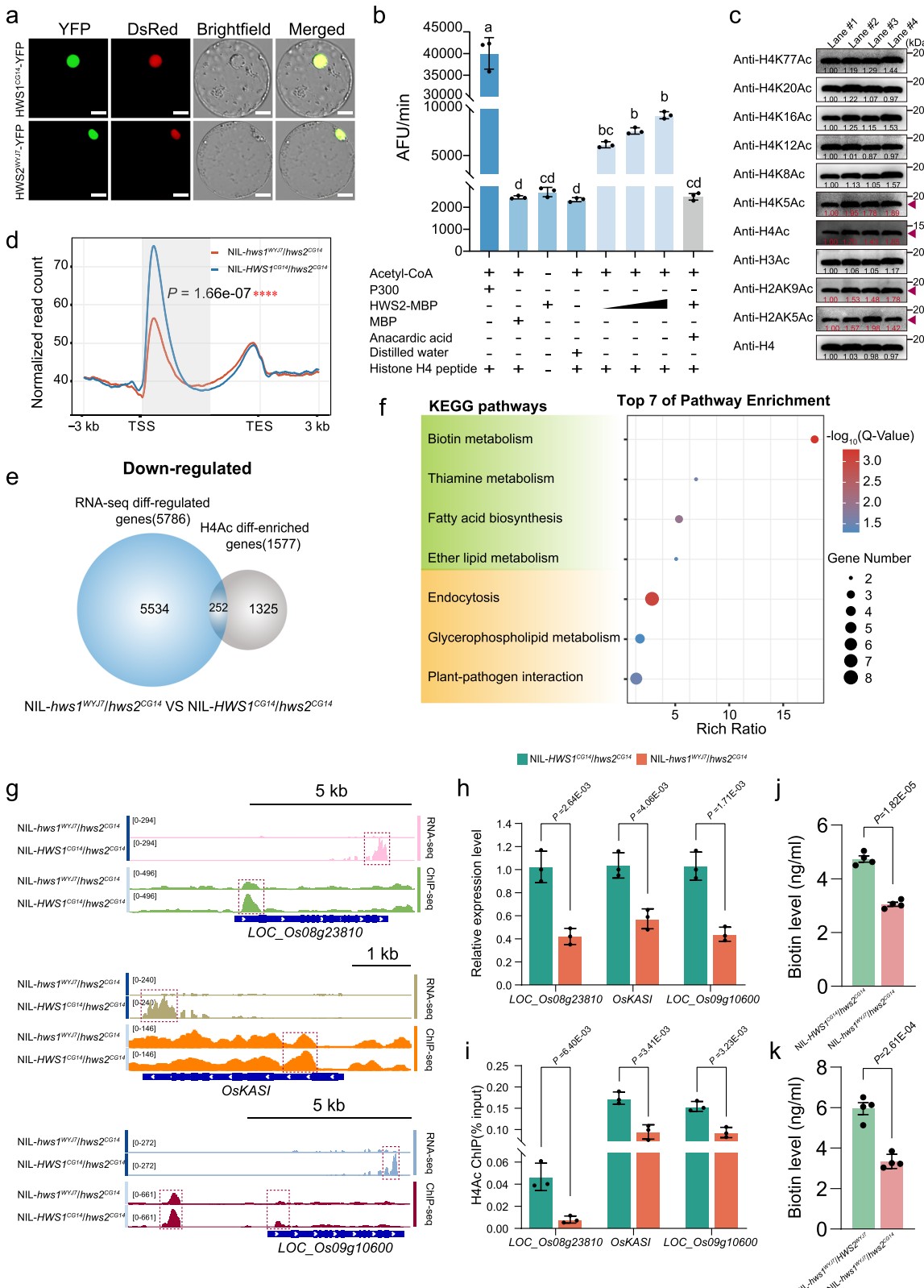

of which were down-regulated and localized at the promoter (88.98%) (Supplementary Fig. 7f). The principal binding site detected in the enriched peaks was the ORB2/RRM-type motif (GGATTTGGGG) (Supplementary Fig. 7g), which basically recruits RNA-binding proteins involved in post-transcriptional regulation.

We hereupon defined the dysregulated genes with H4Ac enrichment peaks at the promoter and/or gene body regions as candidate HWS1/2 target genes (Fig. 3e). Among the categories that overlap with those of the transcriptome datasets, genes related to vitamin metabolism (biotin and thiamine), fatty acid biosynthesis, ether lipid metabolism, and carbohydrate metabolism with reduced H4Ac levels showed altered expression (Fig. 3f and Supplementary Fig. 7h). We, therefore, selected some well-characterized sterility genes belonging to these particular pathways with a strong and uniform reduction in

**Fig. 3 | Involvement of HWS1/2 regulates H4Ac modification and stabilizes transcriptional reprogramming with altered histone acetylation. a** Subcellular localization of HWS1 and HWS2 fused with the yellow fluorescent protein (YFP) reporter. A nuclear localization signal marker (NLS-DsRed) was co-expressed with the HWS1/2-YFP fusion proteins in rice protoplasts. Scale bars = 10 μm. **b** HAT activity testing of HWS2 towards histone H4 by fluorometry. AFU, Arbitrary Fluorescence Units. **c** Endogenous acetylated histone H2A, H3, and H4 levels and substrate specificity determined by western blotting. The specific tested antibodies are shown on the left and the differences in visible band intensity are indicated by red arrowheads. Lane #1: NIL-*hws1^{WYJ7}*/*hws2^{CG14}*; Lane #2: NIL-*HWS1^{CG14}*/*hws2^{CG14}*; Lane #3: NIL-*hws1^{WYJ7}*/*HWS2^{WYJ7}*; and Lane #4: NIL-*HWS1^{CG14}*/*HWS2^{WYJ7}*. **d** The distribution profile of averaged H4Ac occupancy across the gene body in NIL-*HWS1^{CG14}*/*hws2^{CG14}* and NIL-*hws1^{WYJ7}*/*hws2^{CG14}* (****$p < 0.0001$, calculated by two-sided Welch's *t*-test). **e** Venn diagram showing the overlap between RNA-seq differentially-regulated genes with reduced expression levels and ChIP-seq differentially-enriched genes with reduced H4Ac levels. **f** KEGG enrichment bubble plot of the overlapping DEG set in **e**. All significantly enriched (FDR < 0.05) KEGG terms are

shown while the emphasized pathways are additionally indicated with a green background. **g** Genome browser views of mRNA and H4Ac signals of gene examples selected for validation. **h** qRT-RCR expression analysis of two NADH-dependent enoyl-ACP reductase (*LOC_Os08g23810* and *LOC_Os09g10600*) and *OsKASI* (*LOC_Os06g09630*) in the young panicles of NIL-*HWS1^{CG14}*/*hws2^{CG14}* and NIL-*hws1^{WYJ7}*/*hws2^{CG14}* plants. The expression level of each tested gene normalized to rice *Ubiquitin* in NIL-*HWS1^{CG14}*/*hws2^{CG14}* was set to 1.0. **i** ChIP-qPCR analysis showing the relative enrichment of indicated genes identified in **h. j**, **k**, Measurement of biotin level in NIL-*hws1^{WYJ7}*/*hws2^{CG14}* versus NIL-*HWS1^{CG14}*/*hws2^{CG14}* plants (**j**) and NIL-*hws1^{WYJ7}*/*hws2^{CG14}* versus NIL-*hws1^{WYJ7}*/*HWS2^{WYJ7}* plants (**k**). Data in **b**, **h**, **i**, **j**, and **k** are mean ± SEM (*n* = 3 (**b**, **h**, and **i**) or 4 (**j** and **k**) biological replicates). In **b**, different letters denote significant differences ($p < 0.05$, one-way ANOVA with two-sided Tukey's HSD test), *p* values are adjusted and shown in the Source Data file. A two-sided Student's paired *t*-test was used to generate the *p*-values in **h**–**k**. The experiments in **a** and **c** were repeated three times independently, with similar results.

---

sequencing windows to examine their mRNA and H4Ac levels as validation (Fig. 3g)[46]. We examined the expression of these potential genes by qRT-PCR assays, and all of them showed greatly decreased expression in NIL-*hws1^{WYJ7}*/*hws2^{CG14}* plants, which is in line with the results of the transcriptional profile (Fig. 3h). In parallel, chromatin immunoprecipitation quantitative polymerase chain reaction (ChIP-qPCR) results also showed a relatively lower level of histone H4 acetylation in these certain genomic regions (Fig. 3i), implying that in the absence of HWS1/2, the chromatin status is not appropriately established. Besides, determined quantities of biotin were significantly decreased in NIL-*hws1^{WYJ7}*/*hws2^{CG14}* plants in contrast with NIL-*HWS1^{CG14}*/*hws2^{CG14}* (Fig. 3j) and NIL-*hws1^{WYJ7}*/*HWS2^{WYJ7}* (Fig. 3k) plants. Taken together, our findings suggest that HWS1/2 is indispensable for NuA4-dependent H4 acetylation and gene transcriptional activation by proper epigenetic modifications, and is especially essential for gametophyte development through the maintenance of fatty acid, lipid, and vitamin homeostasis.

## Evolutionary trajectory and *EAF6* gene duplication events in *Oryza* species

To trace the evolutionary origins of *HWS1* and *HWS2* (*OsEAF6*), we performed a collinearity analysis around these two distinct loci among some representative *Oryza* species and other distantly related gramineous crop species (Fig. 4a and Supplementary Fig. 8a–d). Comparative genomics showed that there is widespread synteny in the grass family, but that *EAF6* is a single-copy gene in *O.punctata* (BB genome), *O.officinalis* (CC genome), *O.brachyantha* (FF genome) (Fig. 4a), and the majority of AA genome wild progenitors (Supplementary Fig. 8a, b), indicating that the duplication of *EAF6* pairs may have evolved de novo in the *O.sativa* and was distinguished from the other outgroup species *Zea mays* (maize), *Hordeum vulgare* (barley), and *Zizania latifolia* (Manchurian wild rice) (Fig. 4a). These sister groups to *Oryza* mostly showed a common block of homologous synteny around the *EAF6* region retained on chromosome 2 and shared a compatible lineage with their closest wild rice relative (*Leersia perrieri*) aside from the non-overlapping lineage sorting in AA-genome *Oryza* species (Fig. 4a and Supplementary Fig. 8a, b). This suggests that the two copies of *OsEAF6* in *O.sativa* likely originated from their ancestral *EAF6* gene located in a given position of chromosome 2 through speciation, genetic duplication, or horizontal gene transfer. In addition, we made further local collinearity analysis of *HWS1* between *O.sativa* and the other five grasses (Supplementary Fig. 8c) and *HWS2* (Supplementary Fig. 8d) between *O.sativa*, *O.officinalis*, and *L.perrieri*. The results showed that the chromosome segment exhibited significant syntenic relationships from *Zea may* to *Oryza sativa*, but it seemed that *HWS1* came out of nowhere (Supplementary Fig. 8c). On the contrary, the collinearity of the *HWS2* region in related species is much worse,

indicating a possibly huge chromosome structure variation in *HWS2* locus (Supplementary Fig. 8d). Given that transposable elements (TEs) generally cause no synteny[32], we speculated that *HWS1/2* gene pair may be generated by transposon-mediated gene replication. Considering the abundant transposon insertions located nearby, we carried out genome-wide transposon annotation analysis and showed that the segmental genomic duplication carrying *OsEAF6* was probably generated by DNA transposon-mediated gene replication (Supplementary Fig. 9)[47]. To further clarify the evolution process of *HWS1/2*, we constructed the phylogenetic tree and found that the gene tree of the *HWS1* locus was incongruent with the species tree[48] in the certain *HWS1* allele (*hws1^c*) (Fig. 4b). We next generated the gene tree of *HWS1* by using the 133 species (Supplementary Fig. 10a and Supplementary Data 1), and the result showed that all the African rice were more closely to nearly all the *japonica* and some of the *O.rufipogon* rice (Supplementary Fig. 10a).

Although these phylogenetic discrepancies potentially stem from introgression or ILS, coalescence times for regions under ILS should be earlier than the speciation event while introgression occurs later[32] (Fig. 4c). To distinguish which event leads to the incongruence of gene tree and species tree, we first compared the species divergence time (t) with expected coalescence time under ILS ($t_{IL}$) which was estimated based on the gene tree of *HWS1*. Since *O.rufipogon* could be divided into three groups: Or-I/II/III and Or-I is the ancestor of *indica* while Or-III is the ancestor of *japonica*[49], we therefore selected the representative varieties and divided into two major paired-topology categories, W1970 (*HWS1^T*)_W1536 (*hws1^c*)_NH280_W1952 (Fig. 4d) and W1547 (*hws1^c*)_W3012 (*HWS1^T*)_NH280_W1952 (Fig. 4e), both of which belong to Or-I, Or-III, *O.barthii*, and *O.meridionalis*, respectively. The *hws1^c* gene of W1536 (Or-III) and W1547 (Or-I) was more closely related to *HWS1^T* of *O.barthii* and did not match the species tree (Fig. 4d, e). Next, we calculated the synonymous substitution rate (Ks) of 3071 single-copy gene pairs between W1970 (Or-I) and W1536 (Or-III), as well as between W1547 (Or-I) and W3012 (Or-III), and estimated the differentiation time of the two species to be 0.062 and 0.056 MYA, respectively (Fig. 4d, e and Supplementary Fig. 10b, c). Likewise, we calculated the Ks values of the *HWS1* gene pairs between Or-III (*hws1^c*) and *O.barthii* (*HWS1^T*), along with Or-I (*hws1^c*) and *O.barthii* (*HWS1^T*), both of their estimated differentiation time was 0.92 MYA (Fig. 4d, e). The coalescence time of the *HWS1* gene was significantly earlier than the corresponding speciation time (Fig. 4d, e). This strongly suggests that ILS, not introgression, may be probably the major reason for the pervasive signatures of incongruence in these certain *HWS1* alleles (*hws1^c*). To further exclude the introgression event, we conducted the four-taxon D-statistic test with the ABBA-BABA method. No obvious introgression signal was observed in each of the three paired-topology categories ((Or-III, Or-I)_*O.barthii_O.meridionalis*, (Or-II, Or-III)

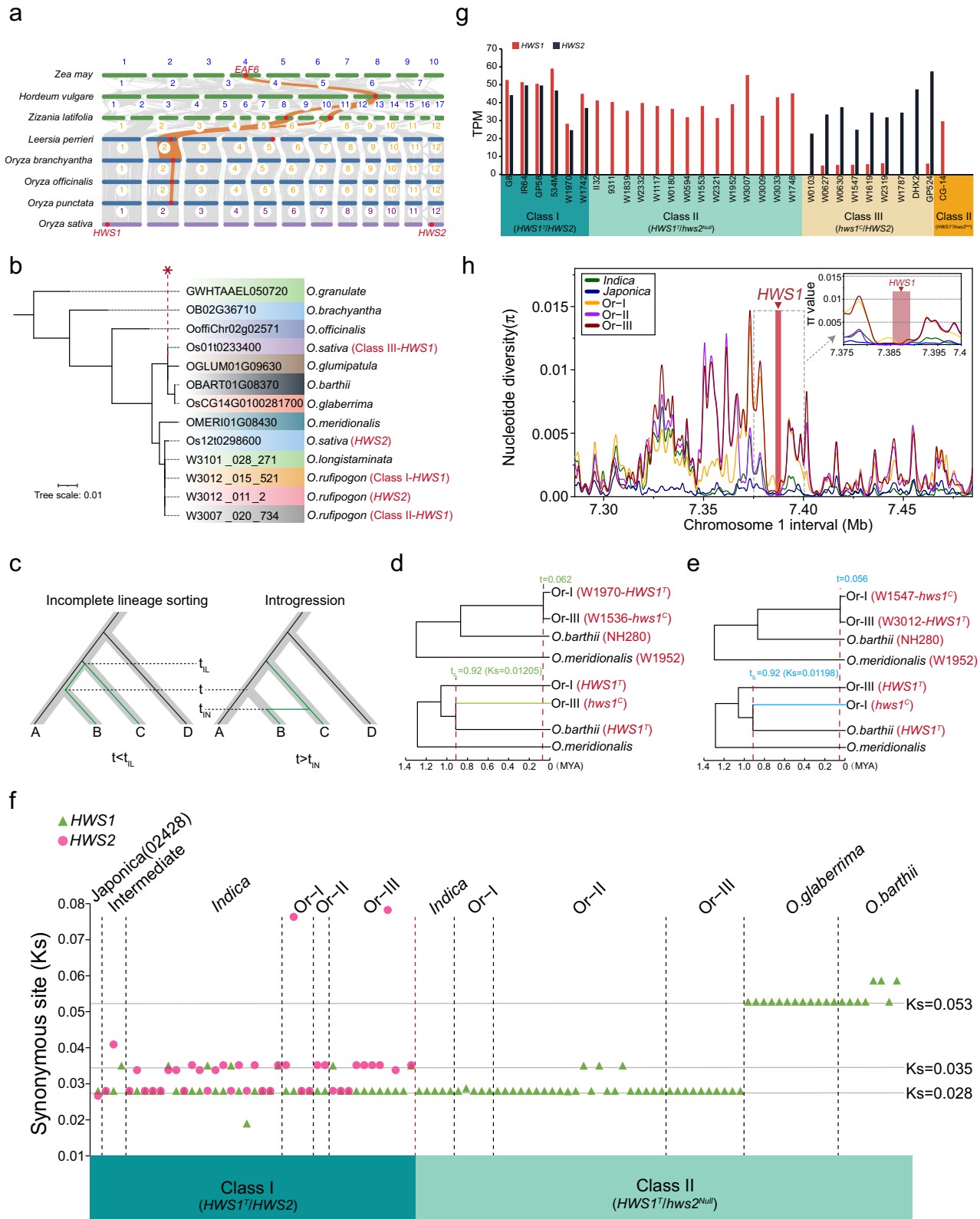

_O.barthii_O.meridionalis,_ and (Or-II, Or-I)_O.barthii_O.meridionalis) in the window where *HWS1* gene was located (Supplementary Fig. 10d).

To comprehensively investigate the evolution and duplication processes of the *HWS1/2* alleles, we selectively compared the genomic sequences among a diverse group of AA-genome rice species. This analysis included wild and domesticated Asian and African rice but excluded a group of certain rice cultivars/accessions due to the presence of multiple tandemly-repeated *HWS2* loci (Supplementary

Fig. 9b and Supplementary Data 1). Based on the pan-genome data, we calculated the synonymous substitution (Ks) values of *HWS1/2* from 133 selected varieties relative to the *HWS1* of *O.meridionalis*. This analysis identified three evolution processes (Ks = 0.028, 0.035, and 0.053, respectively) of the most *HWS1/2* alleles (Fig. 4f). The *HWS1* of *O.rufipogon* and *O.barthii* were diverged from *O.meridionalis* (Ks = 0.028 and 0.053), and then the *HWS2* (*HWS2^T^*) gene in *O.rufipogon* was duplicated from *HWS1* (*HWS1^T^*) (Ks = 0.035) (Fig. 4f), while some of the

**Fig. 4 | Estimation of divergence time and evolutionary scenarios for *HWS1/2* homologs via multiple duplication events. a** Comparative genomes and syntenic gene analysis of *HWS1/2* in rice and selected Gramineae species. *HWS1/2* orthologs are marked with red dots on the chromosomes in each species. Orange lines indicate conserved syntenic blocks harboring *EAF6* (*HWS1/2*). **b** A phylogenetic tree showing the relationships among diploid species of *Oryza* based on *EAF6* sequence variation. The *EAF6* copies from each species are shown with colored backgrounds. Asterisks and green lines represent branch of the gene tree that is inconsistent with the species tree. **c** Diagram shows the distinguishment between ILS and introgression scenarios by using the estimated divergence times. **d, e** The species trees (upper panel) and gene trees (low panel) with divergence time to distinguish between ILS and introgression. Representative varieties of Or-I and Or-III with these two topological structures were selected to estimate the species differentiation time and the differentiation time of *HWS1* gene. The green and blue lines indicate the ILS events. The estimated divergence times of speciation (t) and ILS ($t_{IL}$) are colored in green in Or-III_*O.barthii* topology and blue in Or-I_*O.barthii* topology. **f** Ks distribution of *HWS1/2* in Classes I and Classes II of the 133 selected Asian and African accessions or cultivars relative to the *HWS1* gene on chromosome 1 of *O.meridionalis*. The taxonomic groups are demarcated by dashed lines. **g** Expression analysis of *HWS1* and *HWS2* among the representative rice varieties in the three classes (*n* = 6, 15, and 9 in Classes I, II, and III, respectively) based on RNA-seq data. The *HWS1* and *HWS2* alleles are indicated by red and black bars, respectively. TPM, Transcripts Per kilobase per Million mapped reads. All the TPM values are listed in Supplementary Data 2. **h** A sliding window analysis of π values across a 200 kb chromosomal region including sequences upstream and downstream of *HWS1* in *japonica* and *indica* varieties and three types of wild rice (Or) using published data. The position of the *HWS1* locus is framed with a dotted box and marked with a red arrowhead.

*HWS1* allele (*hws1^C^*) in *O.rufipogon* varieties experienced the ILS event (Ks = 0.049) (Fig. 4c–e and Supplementary Fig. 10b–e). Notably, the transcriptional expression profile further revealed that the *HWS1* genes all underwent gene inactivation resulting from expression failure in certain *O.rufipogon* and *japonica* (*hws1^C^*) (Fig. 4g and Supplementary Data 2). In the three wild rice populations, nucleotide diversity (π) in adjacent regions of *HWS1* gene (7.35 Mb–7.4 Mb) was significantly higher than that of cultivated rice, indicating strong positive selection in this region (Fig. 4h). Evidently, a relatively low π value for the entire 200-kb region was also detected in the *japonica* population (Fig. 4h), indicating the presence of a bottleneck effect shaped by demographic and selective pressure. Nevertheless, the expression failure of *HWS1* in nearly all *japonica* might be due to the founder effect resulting from the domestication of *japonica* rice. Taken together, *HWS1/2*, the functionally redundant paralogs, are thought to be duplicated and originated from a common ancestor in each divergent lineage, and the shared genetic variation between the certain *O.rufipogon* and *O.barthii* may be likely attributed to ILS.

## Variation analysis and functional classification of the *HWS1/2* genes in rice

Using the nucleotide variation identified in the *HWS1/2* gene bodies and the 3′ and 5′ untranslated regions (UTRs), we performed an association test with a local genomic scan and uncovered several mutations that are correlated with gene expression in the same linkage disequilibrium (LD) block (Supplementary Data 2 and Supplementary Data 3). The strongest signal was present at the SNP1_7385301 site, a T-to-C substitution located in the upstream region that is inferred to be a proposed causative variant in *HWS1* (Fig. 5a and Supplementary Fig. 4c). To verify whether the T/C variant in the *HWS1* promoter affects the gene expression, we generated a range of promoter mutation constructs, each fused with *LUC* (*luciferase*) reporter gene and introduced into rice protoplasts to monitor promoter activity (Fig. 5b). The quantitation of firefly luciferase expression showed that the HWS1^CGI4^pro: Luc activity was dramatically reduced after the point mutation ("T" to "C") in SNP1_7385301 site. Instead, the "C" to "T" replacement in *hws1^WYJ7^* rescued the expression failure of *HWS1* in WYJ7, indicating that this core mutation may exert a great impact on the level of promoter activity. Several Asian cultivars including four *japonica* and three *indica* varieties were crossed with SG178 to determine the responsible polymorphic site (Fig. 5c). Plants exhibiting sterility and weakness were observed in the F₂ populations derived from crosses between C-carrying varieties and SG178 (Fig. 5d–g). On the contrary, when carrying the T allele of SNP1_7385301, the functional single nucleotide polymorphism (SNP), all of the descendants with different genotypes showed full seed set (Fig. 5h–j). In summary, our experiments in which we crossed WYJ7 and SG178 with a wide range of rice varieties showed that the targeted variation in *HWS1* accounts for the interspecific sterility in rice. This potentially functional allele is referred to as *HWS1^T^* while the sterile allele lacking

promoter activity or the syntenic segmental block is called *hws1^C^* or *hws2^Null^*, both of which represent a loss-of-function type (Fig. 4g and Supplementary Data 2). Further sequence comparisons revealed that the rice accessions/varieties could be divided into three major *HWS* groups; Class I (*HWS1^T^/HWS2*), Class II (*HWS1^T^/hws2^Null^*), and Class III (*hws1^C^/HWS2*), and all three groups contained *O. rufipogon* accessions (Fig. 5k).

On the basis of the above results, a presumptive model interpreting the duplication and ILS history of *OsEAF6* in *Oryza* genomes was proposed as follows (Fig. 6). The ancestor *EAF6* gene that originated from chromosome 2 of *L. perrieri* was inherited to chromosome 2 in the BB, CC, FF and GG genome species of *Oryza*, and then anchored on chromosome 1 in rice species with AA genome, possibly by transposon-mediated gene replication. This single-copy form of *OsEAF6* gene on chromosome 1 was subsequently retained in the *O.longistaminata*, *O.meridionalis*, and *O.glumipatula* while an interchromosomal copy from chromosome 1 to chromosome 12 was produced in *O.rufipogon* followed by another ILS event occurred between *O.barthii* and *O.rufipogon* (Fig. 6a). In the primitive ancestor gene pool of *O.rufipogon* and *O.barthii*, there existed two major polymorphic *HWS1* alleles: *O.barthii*-like (OB-like; oval) and *O.rufipogon*-like (OR-like; rectangle). Unlike that only OB-like *HWS1* allele was inherited and directed fixed in *O.barthii*, these two types of *HWS1* alleles were both preserved in the ancestor of *O.rufipogon* due to the ILS event. Then, a duplication event that generated the *HWS2* allele replicated from the OR-like *HWS1* and formed the Class I-type *O.rufipogon*. The Class I-type *O.rufipogon* was next crossed with the OB-like wild rice ancestor and formed the heterozygous Class I-type *O.rufipogon*. A random T to C mutation at the promoter region of the OB-like *HWS1* allele was then initiated in the heterozygous Class I-type *O.rufipogon*. This T/C variation in the heterozygous Class I-type *O.rufipogon* eventually gave rise to the form of Class I/II/III-type *O.rufipogon* in the hybrid progenies (Fig. 6b). The various alleles of the *O.rufipogon* accessions eventually evolved to the Class I-III of *O.sativa* and the *O.barthii*, ancestor of African rice, evolved to the Class II type of *O.glaberrima* (Fig. 6).

## Discussion

Gene duplication, a major force that drives gene expansion, occurs spontaneously and exists broadly in RI for species differentiation in allopatric populations[50,51]. Our combined results suggest that the reciprocal loss of redundant genes copied from one another brings about a selective elimination and abortion signal in the *japonica-glaberrima* interspecific recombinants (Fig. 1a, b and Supplementary Fig. 11a). Such biased gamete transmission is accompanied by complete seed and pollen sterility events in the hybrid offspring, which follows the two-locus interaction model but differs from the *S1*-mediated killer–protector system[26–28]. A BLASTN search showed that *HWS2* is completely missing in the African rice gene pool[52] while the nonfunctional allele of *HWS1* is present only in limited *Oryza* species (Supplementary Table 2). The sterile allele *hws1^C^* and the functional

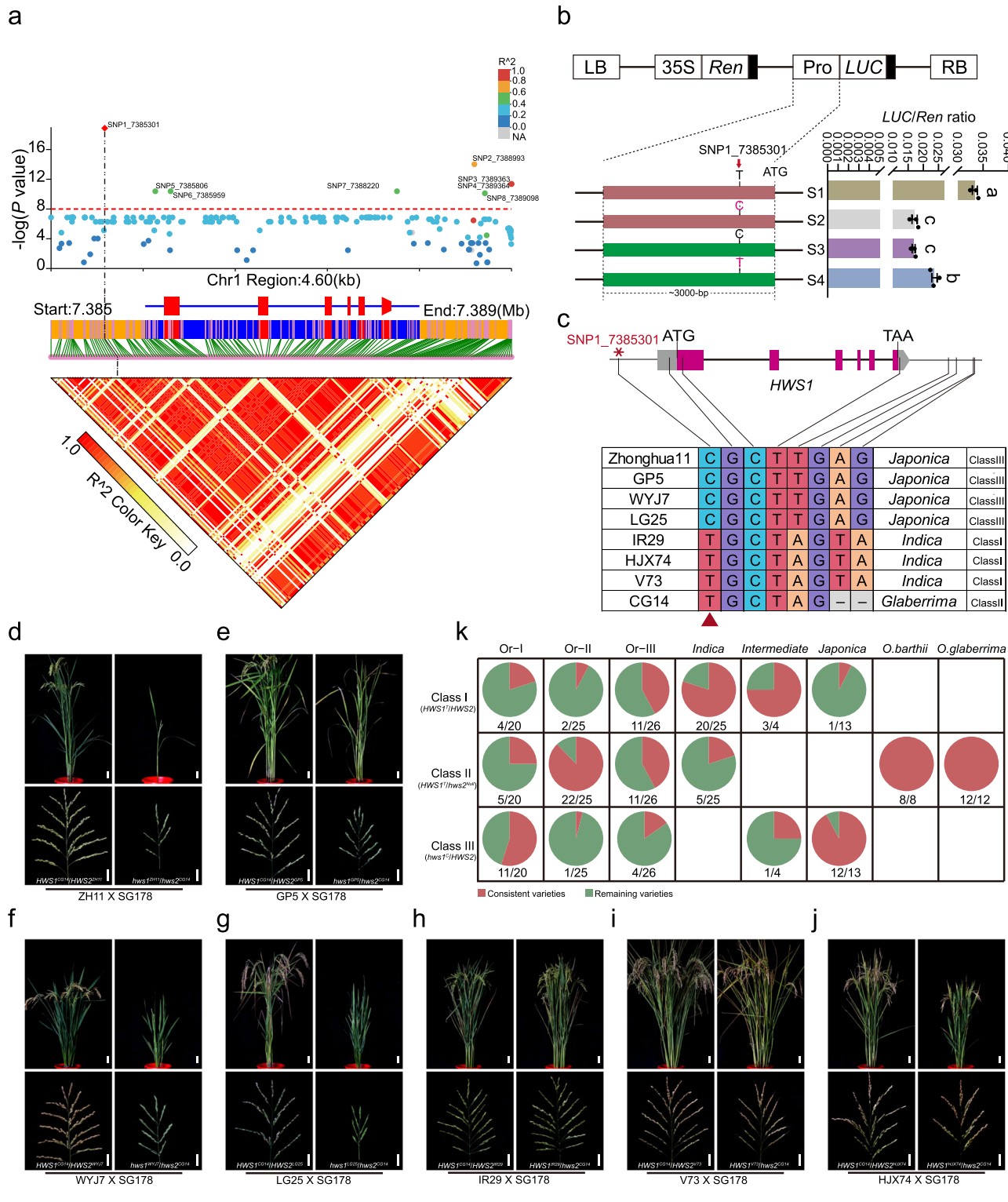

**10**

*HWS2* allele are or near fixed in the *japonica* group, and the functional *HWS1^T* allele is or near fixed in the African rice group, indicating *HWS1/2* may serve as the speciation gene. Additionally, the *HWS2* copy number varies among Asian rice accessions (Supplementary Data 1), suggesting that such genetic model with the parallel divergence lineage is equally applicable to explain the subspecific RI. Unlike previously identified duplicated genes conditioning semi-sterility of rice hybrids: *DPL1/2*[16], *S27/S28*[17], and *DGS1/2*[19], *HWS1/2* serve as indispensable epigenetic modifiers required for cell fate in both reproductive and vegetative organ development, causing developmental defect and complete sterility in the F₂ or the later generation once the

reciprocal gene loss occurs[53]. Despite the fact that the identical recessive genes *hwe1/2* have been reported, the present results have expanded the applicability of the *EAF6*-mediated hybrid sterility model to a wider base of inter-subspecific incompatibility, and revealed that the metabolic disorder of biotin and fatty acid may contribute to the sterility phenotype other than the common autoimmune response in hybrids. Furthermore, we explained the likely lineage divergence process of *HWS1/2* in the *Oryza* genus and phylogenetic conflict at the *HWS1* locus caused by the proposed ILS event, which deepens our understanding of the origins and differentiation of Asian-African rice. None of the existing DNA binding or recognition sites was predicted or

**Fig. 5 | Allelic classification of *HWS1/HWS2* combination and verification of the functional SNP in *HWS1* locus. a** Case/control test for SNPs in the 4.6 kb fine-mapped region around the *HWS1* locus and expression level using a standard one-degree-of-freedom allelic Chi-square test in Plink. Local Manhattan plot (top), gene structure (middle), and LD heatmap (bottom) surrounding *HWS1* are shown. The red diamond in the Manhattan plot shows the position of the proposed causative mutation SNP (SNP1_7385301). *P* values and identified SNPs are presented in Supplementary Data 3. **b** Activity analysis of *HWS1* promoter. Upper part: schematic diagram of the pGREENII 0800-LUC vector fused with *HWS1* promoter. Leftward part: a series of constructs with site-directed mutation *HWS1* promoter. S1, 3000-bp promoter region of *HWS1^{CG14}*; S2, T to C point mutation of S1; S3, 3000-bp promoter region of *hws1^{WYJ7}*; S4, C to T point mutation of S3. Rightward part: expression levels of firefly luciferase normalized by Renilla luciferase. **c** Haplotypes of *HWS1* of various germplasm. The identified SNPs in **a** associated with *HWS1* expression are

arranged in the table (bottom), and the leading SNP variant is marked with a red arrowhead and asterisk. Hybrid combinations between SG178 and seven cultivars including four *japonica* varieties (**d–g**) and three *indica* varieties (**h–j**). The plants in parts **d–j** shown in the right panels exhibited typical sterile and dwarf phenotypes compared with the F₂ control individuals shown on the left. Scale bars = 5 cm (top), 5 cm (bottom). **k** Allele frequencies of the causal polymorphisms in *HWS1/HWS2* in the three groups comprising 133 AA-genome *Oryza* accessions. The total number and proportion of accessions in each class and population are given beneath each pie chart. The blank grid indicates that there were no varieties in this Class. All rice accessions are listed in Supplementary Data 1. Data in **b** are means ± SEM (*n* = 3 biological replicates). Different letters denote significant differences (*p* < 0.05, one-way ANOVA with two-sided Tukey's HSD test). *P* values for **b** are adjusted and shown in the Source Data file.

detected in the *HWS1* promoter region within this core mutation (SNP1_7385301) (Supplementary Fig. 4c), which suggests that the resulting absence of or lack of expression of *HWS1* could be due to chromatin remodeling. It is tempting to speculate that gene loss of *HWS2* (*hws2^{Null}*) or gene degeneration caused by expression inactivation (*hws1^c*) in each copy of a duplicated pair is likely to be neutral and will not have adaptive consequences for organisms.

Both ILS and introgression can result in identical genotypes across species that are scattered throughout phylogenetic trees and are independent of speciation order. However, researchers tend to focus more on introgression rather than ILS. Ignoring ILS might contribute to incorrect interpretations of morphological evolution in different lineages that undergo speciation events in rapid succession[32]. *HWS1/2* is most likely to be viewed as a prospective marker gene to deduce the evolutionary history in the light of its allelic distribution and polymorphism among all *Oryza* species. Our analysis of genomic data showed evidence for ILS that occurred during African and Asian rice speciation (Fig. 4c–e and Supplementary Fig. 10b–d), which certainly expands our knowledge of the evolution and hybridization history of Asian and African rice. It is, therefore, of particular interest that *HWS1/2* may serve as unique ILS loci to counterbalance the genetic diversity and break through the innate reproductive barrier in rice.

These two atypical selfish genetic elements both constitute parts of the multi-subunit NuA4 complex that modulates chromatin accessibility at regulatory regions and the assembly of transcriptional machinery for male-female gametophyte development, presumably via transcriptional-repressive epigenetic modification rather than being toxic to kill gametes, leading to decisive sterility heredity in rice (Fig. 3b–d and Supplementary Fig. 11b). By analogy to house-keeping genes, *HWS1/2* are likewise constitutively expressed at relatively high levels (Supplementary Fig. 4a, b), suggesting that they play indispensable roles in determining cell fate. The Cas9-induced loss-of-function mutants generated in this study provide experimental evidence supporting a central hub for the two genes in both reproductive development and growth regulation (Fig. 1g, h). Previous studies in plants showed that autoimmune responses associated with *NB-LRR* disease-resistance genes or *R* gene are major and common causes of hybrid weakness and breakdown[54,55]. However, we did not observe any visible lesions and typical programmed cell death (PCD) induced in the leaf and basal nodes of sterile plants compared to fertile plants (Supplementary Fig. 12a–f). No obvious upregulated or downregulated DEGs involved in autoimmune or defense response were enriched in the RNA-seq or ChIP-seq data, suggesting that this weak phenotype may not result from autoimmunity (Fig. 3f and Supplementary Fig. 12g).

From analyses of multi-omics data (Fig. 3f and Supplementary Fig. 7c, h), it can be inferred that genes involved in the remaining metabolic processes (e.g., sugar, carotenoid, and flavonoid synthesis), Golgi-related cellular components, and endocytosis may be the other likely HWS1/2-dependent downstream regulators, leading to male/

female sterility due to nutrient deficiency and autotrophic collapse. In addition, the overlap between H4Ac-enriched genes and transcriptional DEGs only occupied a small proportion, suggesting that EAF6 may also play a part in acetylating nucleosomal histone H2A or H3 as well for transcriptional activation (Fig. 3e). Overall, a comprehensive understanding of the *HWS1/2*-mediated reproductive isolation system not only broadens our knowledge concerning the genetic composition of *Oryza* species, but it also promotes the instructive utilization of gene exchange in inter-subspecific genetic improvement.

Transformation experiments demonstrated that overexpression of the *HWS2* gene did not result in a weakness syndrome, and the plants possessed normal fertilization capability (Supplementary Fig. 13), indicating that there are no obvious signs of negative effects from the number of *OsEAF6*-specific transcripts. As an alternative approach to overcome interspecific and subspecies reproductive isolation caused by the loss of *HWS* gene copy number in rice, artificially engineering *HWS1/2* loci provides another potential, effective target to restore pollen and spikelet fertility together that can be used in distant hybridization in rice breeding. This may require us to fully consider the practical feasibility of natural hybrid-compatible varieties as founders for use in inter-subspecific breeding other than general gene editing[56]; an example is '02428', a *japonica*-type widely compatible cultivar with a well-preserved function of *HWS1* and *HWS2* (Supplementary Data 1).

## Methods
### Plant materials and growth conditions
The CSSL library ('SG') was constructed from a cross between CG14 (*O.glaberrima* Steud.) as the donor parent and 'Wuyunjing7' (WYJ7; *O.sativa* L. ssp. *japonica*) as the recurrent parent through multiple backcrosses. A selected line, SG178, which carries substituted chromosomal segments containing the *HWS1* and *HWS2* loci from African rice, was used to develop segregating populations for high-resolution mapping of *HWS1/2*. NIL(*HWS1*) and NIL(*HWS2*), two lines that harbor an almost consistent genetic underpinning, can only be maintained in a heterozygous state at the respective *HWS1* or *HWS2* locus in the WYJ7 background, and the genotypes of each NIL were determined by the whole genome-covering markers. The self-pollinated progeny of NIL-*HWS1/HWS2* (*HWS1* and *HWS2* heterozygotes, W1G1|W2G2 genotype) was used for the phenotypic investigation of RI syndromes. All field-cultivated materials were grown in a managed planting base at two locations in China; Shanghai in the summer and Sanya in the winter.

### Map-based cloning of *HWS1* and *HWS2*
The segregating BC₅F₂ population derived from the SG178 × WYJ7 F₁ hybrid was developed for fine-mapping of *HWS1* and *HWS2* simultaneously, and the seed fertility or segregation ratios of RI characters in all tested plants were calculated as an appraisal index for QTL linkage analysis. The *HWS1* locus was roughly delimited to a genomic interval on chromosome 1 between marker loci ID1-2 and ID1-3 using 6,570 BC₅F₂ individuals. A total of 38,580 BC₅F₂ plants were then further

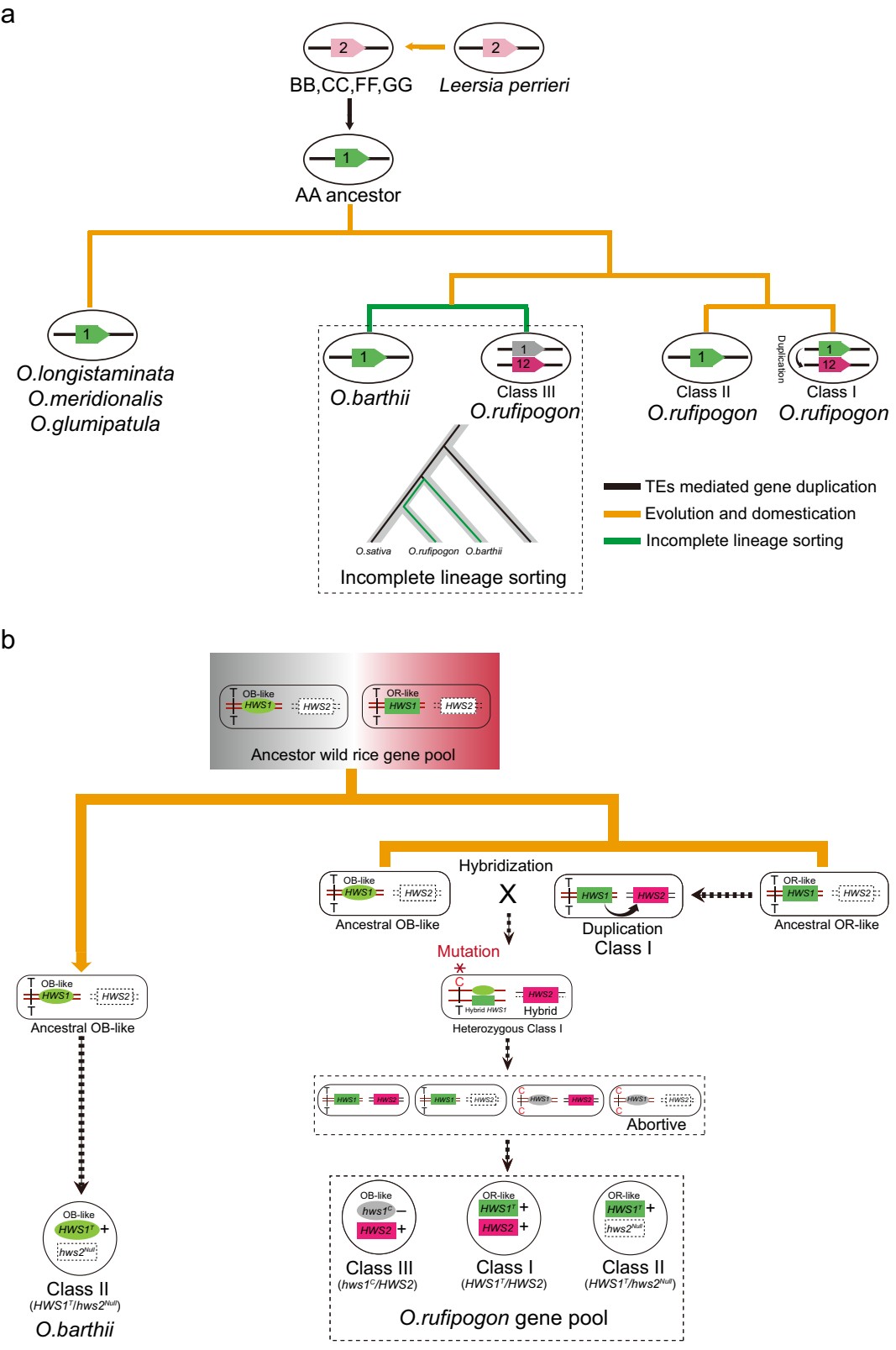

screened using five flanking markers to detect genetic recombination. Using newly developed molecular markers, the location of *HWS1* was finally narrowed down to a 35.25-kb region embodying three annotated genes. The sketchy location of *HWS2* was first detected on chromosome 12 and ultimately mapped to a 337.5-kb region between marker loci ID12-8 and ID12-9 around the centromere using 60,481 BC$_5$F$_2$ plants. The query of gene fragments was searched by a web-

based tool (http://ricerc.sicau.edu.cn/). All primers used for genotyping are given in Supplementary Data 4.

### Evaluation of pollen and spikelet fertility
To measure the viability of male gametes, pollen grains released from broken anthers of each single spikelet at the heading stage were mixed and stained with 1% (w/v) iodine-potassium iodide (I$_2$-KI) solution to

**Fig. 6 | Proposed evolutionary and ILS history and lineage model of the *HWS1/2* genes in *Oryza* gene pools. a** a simplified schematic diagram illustrating the evolutionary trajectory of the *OsEAF6* genes after gene duplication and ILS events in AA and non-AA genome *Oryza* species. The currently existing *OsEAF6* alleles in the *Oryza* genus trace back to the progenitor singleton on chromosome 2 of *L. perrieri* and evolved into chromosome 2 of BB, CC, FF, and GG genome rice. Finally, the *OsEAF6* gene replicated to chromosome 1 of the ancestor AA genome rice. The orange and green lines indicate the evolution and ILS events, and the black arrows indicate the duplication event. The numbers 1, 2, and 12 in each oval represent the chromosomes on which the copied *EAF6* genes are located. **b** a diagram showing the origin and generating process of Class I-III of *HWS1/2* combination in *O.barthii* and *O.rufipogon*. Two *HWS1* types, *O.barthii*-like (OB-like; oval and light green) and *O.rufipogon*-like (OR-like; rectangle and dark green) alleles, co-existed in the ancestral gene pool of *O.rufipogon* and *O.barthii* while *HWS2* was generated later by duplication from the OR-like *HWS1*, leading to the generation of Class I-type *O.rufipogon*. By crossing with the OB-like ancestor wild rice, the heterozygous Class I-type *O.rufipogon* was subsequently formed. The produced T-to-C mutation in the promoter region of OB-like *HWS1* allele occurred in the heterozygous Class I-type of *O.rufipogon*. The heterozygous Class I-type of *O.rufipogon* carrying the T/C variation in the OB-like *HWS1* eventually produced the Class I-III of *O.rufipogon* in the self-progeny. mRNA transcripts of *HWS1* and *HWS2* are marked with the characters "+", and "−" in rectangles and ovals to indicate the transcriptional products with high accumulation (*HWS1*, dark and light green; *HWS2*, purple) or transcripts that are expressed at low levels (*hws1*, gray) based on the speculative causative mutation. Additionally, the absence of *HWS2* is signified by dashed lines, and the hybrid *HWS1/2* is indicated with the solid and dashed lines.

determine pollen fertility using a light microscope. Dark-stained pollen grains with a nearly spherical shape were scored as fertile, whereas unstained or partially stained yellow-brown and misshapen grains were deemed as sterile. The seed-setting rate on the main panicle was defined as a quantitative indicator to evaluate spikelet fertility after maturity.

### Cytological analyses of anther and embryo sac development

Florets from fresh spikelets at various stages of development were excised as samples and were fixed in FAA solution (50% ethanol, 37% formalin, and 100% acetic acid). Dehydrated samples were then embedded in resin and sectioned with an EM UC7 Ultramicrotome (Leica). The semi-thin sections were subsequently stained with fast green and observed under an AxioCam MRc light microscope (Zeiss). Microspore development is roughly classified into 14 stages based on the length of the flower glume in rice and previous descriptions[57,58]. For cross-section and TEM analysis, anthers from the corresponding stages were collected and then fixed in 2.5% (v/v) glutaraldehyde solution for at least 24 h and treated as described by Guo[59]. Images were captured with a Hitachi H7650 electron microscope using a voltage of 80 kV. SEM was performed and modified accordingly, referring to a previous study[60]. Ripe anthers and exposed mature pollen grains were detached, frozen in liquid nitrogen, and freeze dried. The lyophilized specimens were placed in copper tables and visualized by a field emission electron microscope (Zeiss Merlin Compact).

To examine the structure of the embryo sacs, the samples were prepared using a previously published procedure[61]. Before staining, the fixed samples were washed again and transferred to 70% (v/v) ethanol. The lemma and palea were removed and discarded to take out the whole ovary. Soon afterward, the tissues were rehydrated through an ethanol series and transferred to distilled water. All ovaries were incubated in 2% (w/v) aluminum potassium sulfate dodecahydrate (Aladdin, A103914) for 10-20 min, followed by staining in 10 mg/L Eosin-B (Aladdin, E100232) for 10-12 h at room temperature. Each medium-stained ovary was repeatedly dyed in 2% (w/v) aluminum potassium sulfate dodecahydrate once again for an additional 8 h and was then processed through a graded alcohol series. Finally, the tissues were cleared in 50% (v/v) methyl salicylate (Macklin, M813577) for 1–2 h and immersed in pure methyl salicylate for 1 h. The viability of the embryo sacs was ascertained by examination using a Leica TSC SP8 STED confocal laser microscope.

### Plasmid construction and plant transformation

To construct the gene overexpression construct, the full-length coding sequence of *HWS2^{WYJ7}* was amplified using cDNA from WYJ7 leaves and was then inserted into the binary vector pHB downstream of two tandemly repeated CaMV35S promoters with an EGFP tag. The functionally inactive mutants were all generated by the CRISPR/Cas9 gene-editing system, and the selected sequences of the designed target sites were synthesized, annealed, and cloned into the pYLCRISPR/Cas9 vector using the *BsaI* site[62]. The primer sets and single guide RNA (sgRNA) sequences are given in Supplementary Data 4. All rice genetic transformation experiments were performed using *Agrobacterium*-mediated transformation of WYJ7, SG178, and/or NIL-*HWS1^{CG14}/HWS2^{WYJ7}* callus tissue with the *A. tumefaciens* strain EHA105. Phenotypic measurements were continuously taken in each generation of transgenic plants.

### RNA extraction and gene expression analysis

Anthers at various developmental stages were harvested separately based on spikelet morphology and growth position of the floret[57,58]. Total RNA was extracted from the rice tissues/organs and was purified using the E.Z.N.A. Total RNA Kit (Omega), and a given mass of each RNA sample were then incubated with DnaseI for cDNA synthesis using the PrimeScript™ RT reagent kit with gDNA Eraser (Takara) following the manufacturer's instructions. Quantitative real-time PCR (qRT-PCR) assays were performed and analyzed using the QIAquant 96 5plex Real-Time PCR System (Qiagen) in a reaction mixture containing Fast Start Universal SYBR Green Master Mix with ROX (Roche), and there were three biological replicates for each sample. The rice *Ubiquitin* gene was used as the internal control to normalize the gene expression data. The relevant primer pairs used for gene amplification are given in Supplementary Data 4.

### Subcellular localization assay

The *35S::HWS1/2-YFP* constructs were produced separately and introduced into isolated rice protoplasts using a PEG-mediated plasmid uptake method. A fluorescently labeled organelle marker containing a nuclear localization signal (NLS) fused with DsRed was co-transfected as an indicator. Confocal imaging analysis was performed using a confocal laser scanning microscope (Zeiss LSM880) at excitation wavelengths of 514 and 563 nm. Primers used in this experiment are given in Supplementary Data 4.

### Protein purification and histone acetyltransferase activity assay

The *HWS2^{WYJ7}* coding sequence was PCR amplified and inserted between the *BamHI/EcoRI* restriction sites of pMAL-c5x to produce the MBP-HWS2^{WYJ7} expression vector by homologous recombination. All constructs were then transformed into *Escherichia coli* BL21 (DE3) pLysS Rosetta-gami 2 strain, and the recombinant fusion proteins were induced by the addition of 0.1 mM isopropyl-β-D-1-thiogalactopyranoside (IPTG) and prepared using the AKTA Avant protein purification system. The relevant primer sequences are given in Supplementary Data 4. For the in vitro HAT assay, the intrinsic HAT activity was detected using the fluorescent HAT Assay Kit (Active Motif) following the manufacturer's instructions with some modifications. In brief, the purified proteins (MBP-HWS2, MBP, and positive control p300) were incubated with 20 μl histone H4 peptide in a 50 μl reaction mix that included 6 μl of 5× HAT assay buffer and 5 μl of acetyl-CoA (0.5 mM) at 28°C for 1 h. The corresponding fluorescence signals were detected using a spectrophotometer with excitation at 375 nm and emission at 460 nm (Thermo Scientific, Varioskan LUX).

## High-throughput SNP genotyping

DNA samples isolated from CG14, WYJ7, and SG178 were processed and analyzed by Shanghai OE Biotech Co., Ltd. (Shanghai, China). Genotypic data for genome-wide scanning was generated using the Affymetrix SNP chip to track the DNA segments from the parental genomes. The experimental procedures and analytical processes are summarized as follows: fresh rice leaves were immediately frozen in liquid nitrogen and genomic DNA was extracted using the DNeasy Plant Mini Kit (Qiagen). PCR-amplified DNA fragments with high quality and appropriate concentration were then marked and hybridized with Axiom Rice 44k array after random interruption and purification. Non-specific binding was removed by multiple washing and each nucleotide site was finally detected to generate a raw genotype file. Among all the 42,961 SNP probes on the genotyping array, the number of markers filtered to be homozygous and polymorphic SNPs between the two parental lines, CG14 and WYJ7, was 1,233, distributed across the 12 chromosomes.

## Whole-genome resequencing

The genomes of SG178 and WYJ7 were resequenced on the Illumina HiSeq4000 platform by Novogene Bioinformatics Technology Co., Ltd. (Beijing, China). The young seedlings were first harvested to extract DNA, and sequencing libraries were prepared by sonication and ligation with adaptors. The PCR-amplified libraries were purified and sequenced to obtain the raw data with quality control. The clean sequence reads from each sample were subsequently mapped to the 'Nipponbare' reference genome (IRGSP-1.0) (https://rapdb.dna.affrc.go.jp/). All genetic variants were identified with functional annotation by comparison to 'Nipponbare' (*O.sativa* ssp. *japonica*).

## RNA-seq analysis

RNA deep sequencing and analysis were performed by BGI-Tech (Shenzhen, China). Total RNA for each three biological replicates consisted of panicles (approximately 10 cm in length, 5 individual plants per biological replicate) pooled from NIL-*hws1*$^{WYJ7}$/*hws2*$^{CG14}$ and NIL-*HWS1*$^{CG14}$/*hws2*$^{CG14}$ that were isolated independently after flash freezing in liquid nitrogen. The mRNA fraction was enriched using oligo (dT) magnetic beads and was uniformly fragmented to a specified length. First-strand cDNA was generated using random hexamer-primed reverse transcription, followed by second-strand synthesis. The cDNA was next subjected to end-repair and the ends were 3'adenylated. Adapters were then ligated to the ends of 3'adenylated cDNA fragments. The cDNA fragments with adapters were then amplified and purified with Ampure XP Beads (Agencourt), and were dissolved in EB solution. The quality of the mRNA-seq libraries was determined using the Agilent Technologies 2100 bioanalyzer, and the libraries were amplified with phi29 to make DNA nanoballs (DNBs) possessing more than 300 copies per molecule. The DNBs were loaded into the patterned nanoarray and the single-end 50 (pair-end 100/150) base reads were generated using the combinatorial Probe-Anchor Synthesis (cPAS) method. RNA-seq reads were aligned to the rice reference genome (MSU-7.0) (http://rice.uga.edu/index.shtml) using Hisat2 (v2.05). Transcription levels of tested genes were quantified by FPKM (fragments per kilobase of exon per million mapped reads). The criteria of fold-change ≥2 and *q*-value ≤ 0.05 were established to identify the differentially expressed genes.

## ChIP-seq analysis

Data acquisition and handling were performed by Jiayin BioTech Ltd. (Hangzhou, China). The tissues used for RNA-seq were harvested synchronously at the same developmental stage as the samples used in the ChIP-seq experiment with 3 individual plants but without biological repeat. The fresh immature panicles were first cross-linked in 1% formaldehyde (Sigma, 47608) under vacuum for 20 min. The nuclear chromatin was isolated by sucrose gradient centrifugation and then sonicated. The lysate was further diluted with ChIP buffer and immunoprecipitated using an anti-histone H4Ac antibody (Millipore, 06-866). The ChIP-DNA complex was then extracted and purified using the Universal DNA Purification Kit (#DP214). The ChIP-seq library was prepared using a ChIP-seq DNA sample preparation kit (NEBNext® Ultra$^{TM}$II DNA) as directed by the manufacturer. The extracted ChIP-DNA fragments were ligated to specific adaptors and sequenced on an Illumina Novaseq 6000 instrument. The sequence reads were aligned to the rice reference genome (MSU-7.0) (http://rice.uga.edu/index.shtml) and the uniquely mapped reads used for peak calling were analyzed using MACS2 (version 2.1.4) peak caller software. Motif analysis was conducted using HOMER findMotifsGenome.pl tool and the peaks were annotated using the annotatePeak function in ChIPseeker (version 1.26.2). Analyses of the differences between the two samples were performed using the ChIPDiff (version 2.3.4) program, and the detailed read count data was loaded into the IGV (version 2.8.9) genome browser for visualization.

## Native ChIP-qPCR assay

It was by analogy that the young inflorescences of the two NILs (NIL-*hws1*$^{WYJ7}$/*hws2*$^{CG14}$ and NIL-*HWS1*$^{CG14}$/*hws2*$^{CG14}$) were sampled and fixed in a crosslinking solution (10 mM Tris-HCl [pH 8.0], 400 mM sucrose, 1 mM PMSF, 5 mM β-mercaptoethanol, and 1% formaldehyde) under vacuum for 20 min at room temperature. Chromatin isolation was performed using the EZ-ChIP kit (Merck) using the standard protocol with minor modifications. Briefly, the soluble chromatin was incubated with an antibody directed against Histone H4Ac and was then precipitated with Protein A/G at 4 °C overnight. The immunoprecipitated complex was retrieved and analyzed by qRT-PCR using region-specific primers, and the data were normalized by input DNA. The sequences of the primers used for ChIP-qPCR can be found in Supplementary Data 4.

## Protein extraction and western blotting

Nucleoproteins were extracted from 10 cm panicles of NIL-*hws1*$^{WYJ7}$/*hws2*$^{CG14}$, NIL-*HWS1*$^{CG14}$/*hws2*$^{CG14}$, NIL-*hws1*$^{WYJ7}$/*HWS2*$^{WYJ7}$, and NIL-*HWS1*$^{CG14}$/*HWS2*$^{WYJ7}$ plants using the Nuclear isolation kit (CELLYTPN-1KT, Sigma-Aldrich), and all steps were done as indicated in the instruction and operation manual. Protein extracts were boiled and terminated in SDS sample buffer, the proteins separated by denaturing polyacrylamide gel electrophoresis (SDS-PAGE) were then transferred to polyvinylidene difluoride (PVDF) membranes (Amersham, GE Healthcare). After blocking with 5% skim milk in TBST buffer (20 mM Tris/HCl, pH 7.6, 137 mM NaCl, 0.1% Tween) at 4°C overnight, the membranes were subsequently immunoblotted with primary antibodies using recommenced dilutions, followed by washing and incubation with a similarly diluted secondary antibody (goat anti-rabbit, Agrisera). The protein bands were then imaged using a UVP ChemStudio PLUS (Analytic Jena) after incubation in prepared ECL solutions (Amersham, GE Healthcare) for 3-5 min at room temperature in the dark.

## Transient expression assays of promoter activity

The ~3000-bp, promoter fragments upstream of the *HWS1*$^{CG14}$ and *hws1*$^{WYJ7}$ translation start site were amplified from CG14 and WYJ7, respectively, and the fragments with site-directed mutation at SNP1_7385301 site were synthesized by Shanghai Xitubio Biotechnology. All of these fragments were inserted into pGREENII 0800-*LUC* vector. The rice protoplast was isolated from two-week-old rice seedling leaves after sowing. Each of the *HWS1* promoter-*LUC* gene fusion constructs was transfected into rice protoplasts by PEG-mediated transformation. *LUC* and *REN* luciferase signals were measured using the Dual-Luciferase Reporter Assay System (Promega).

## Measurement of biotin levels

The biotin levels were measured by using a commercial Enzyme-Linked Immunosorbent Assay (ELISA) kit (E-IR-R501, Elabscience). In general,

50 µl testing sample was added into each well of the ELISA plate and then incubated with 50 µl Avidin-HRP working solution for 30 min at 37°C. After 3 rounds of wash, the substrate regent (90 µl each well) was added and then incubated for 15 min at 37°C in the dark. Finally, the optical density values of samples at the 450 nm wavelength (OD450) were recorded immediately after the addition of 50 µl stop solution. The biotin levels in different samples were calculated by comparing the OD450 values of the samples to a standard four-parameter logistic curve by using the ELISACalcu (version 0.1) software.

## In situ hybridization

Pollens of NIL-$hws1^{WYJ7}$/$hws2^{CG14}$ and NIL-$HWS1^{CG14}$/$hws2^{CG14}$ at different development stages were used for investigating the expression pattern of *HWS1* in pollen grains. The DIG-labeled probe was used for detecting *HWS1* expression and the sense probe for *HWS1* was used as the negative control. All probes were synthesized by Gefan Biotechnology Company (Shanghai, China). Samples were fixed in 4% paraformaldehyde overnight and then incubated for hybridization with the detection probes at 37 °C for 24 h according to standard procedures[63]. The DIG-labeled probe for *HWS1* and the negative sense probe are listed in Supplementary Data 4.

## Antibodies used in this study

The following rabbit antibodies were used for the western blotting experiment, primary antibodies: anti-Histone H4 (Abcam, ab177840, rabbit monoclonal), anti-Histone H4Ac (Millipore, 06-866, rabbit monoclonal), anti-Histone H4K5Ac (Abcam, ab51997, rabbit monoclonal), anti-Histone H4K8Ac (Abcam, ab45166, rabbit monoclonal), anti-Histone H4K16Ac (Abcam, ab109463, rabbit monoclonal), anti-Histone H4K20Ac (Abcam, ab177188, rabbit monoclonal), anti-Histone H4K77Ac (Abcam, ab241117, rabbit polyclonal), anti-Histone H3Ac (Active Motif, 39040, rabbit polyclonal), anti-Histone H2AK5Ac (Active Motif, 39108, rabbit polyclonal), and anti-Histone H2AK9Ac (Active Motif, 39110, rabbit polyclonal). Secondary antibody: goat-anti-rabbit IgG (H + L) (Proteintech, SA00001-2). A dilution of 1:2000 is used for primary and secondary antibodies in the western blotting experiment. For ChIP-seq and ChIP-qPCR experiment: anti-Histone H4Ac (Millipore, 06-866, rabbit monoclonal) is used, with a dilution of 1:200.

## Identification and phylogenetic analysis of *EAF6* genes in grass family

The genomes of *Zea mays*, *Hordeum vulgare* (Version TRITEX), *Leersia perrieri*, *Oryza brachyantha* (FF genome), *Oryza punctata* (BB genome), and *Oryza barthii* were all retrieved from ftp://ftp.gramene.org/pub/gramene/release-65/. The genomes of *Zizania latifolia* and *Oryza officinalis* (CC genome) were downloaded from the NCBI (GenBank accessions GCA_000418225.1 and GCA_008326285.1, respectively). The genome of *Oryza granulata* (GG genome) was obtained from the National Genomics Data Center under accession number GWHAAEL00000000. Publicly available data for the genomes of 20 African rice accessions from the rice super pan-genome were obtained from the Genome Warehouse (GWH) (http://bigd.big.ac.cn/gwh/) under PRJCA004295. The genomes of 71 wild rice accessions and 12 cultivars were from our recent de novo assemblies.

To retrieve the orthologs of the *EAF6* gene in the representative gramineous plants, the coding sequences of *HWS1* and *HWS2* were inputted into the database as queries to extensively search the matched hits based on a BLAST approach (version 2.2.28)[64]. All identified *EAF6* genes were included in an all-versus-all alignment using MAFFT (version 7.158b) with the default parameter settings[65]. A maximum likelihood phylogenetic tree was reconstructed using IQ-TREE (version 1.6.3) with 1000 bootstrap replicates[66].

The 3071 single-copy gene pairs between W1970 and W1536, as well as between W1547 and W3012 were obtained using OrthoFinder v2.3.8[67]. The synonymous substitution ratio (Ks) of the gene pairs was calculated using ParaAT2.0 software[68]. The peak Ks values were converted to divergence time according to the formula T = Ks/2λ (T, time; λ, average substitution rate) by using an average substitution rate of 6.5 E−09[69] for grasses to infer speciation. The Ks values for the *HWS1* gene that underwent incomplete lineage sorting in W1536 and W1547 and the corresponding gene in *O.barthii* were also calculated by ParaAT2.0 software, and their differentiation time was calculated by the above formula.

## Gene expression analysis

Data fetching and aggregate of RNA sequencing results from 33 rice varieties were downloaded from the in-house RNA-seq pipeline (BioProject number: PRJCA002103), National Genomics Data Center (http://bigd.big.ac.cn/). For transcript quantification, transcripts per million (TPM) values were determined using the Salmon method to normalize the gene expression level in the RNA pools[70]. The other available transcriptomic profiles of 71 wild rice accessions and 12 African cultivars were additionally obtained from our recent de novo assemblies and the same optimization method was applied to them.

## Repetitive sequence annotation

The whole-genome repetitive sequence analysis applied to all 33 rice cultivars is accessible at 'Rice Resource Center' (http://ricerc.sicau.edu.cn/). Transposons and tandem repeats from *L. perrieri*, *O. officinalis*, and *O. punctata* were annotated in detail with the Extensive de novo TE Annotator (EDTA, version1.9.6) using the curated TE library (rice 6.9.5.liban) with the parameters '-overwrite 0 -sensitive 1 -anno 1 -evaluate 0'[71].

## Genomic comparison and visualizations

MCScan software (Python version) was used to scan multiple genomes, run the pairwise synteny search program, and align putative homologous chromosomal regions with the LAST results of *O. sativa* versus the other selected grass species[72]. The genome pairwise synteny visualization was generated using the command 'Python -m jcvi.graphics.karyotype' while microsynteny visualization was produced using the command 'Python -m jcvi.graphics.synteny'.

## Gene duplication analysis

Based on the pan-genome data, we calculated the synonymous substitution (Ks) values of *HWS1/2* from selected varieties relative to the *HWS1* of *O.meridionalis* (a more ancient wild rice variety to the AA genome rice species) using ParaAT2.0 software with the parameter '-m clustalw2 -f axt -g -k'. The Ks distribution of paired *HWS1* and *HWS2* genes on orthologous data sets from 133 selected rice varieties was used as a rough indicator to determine gene duplication events and the numeric value was negatively correlated with divergence time and kinship with its ancestral copy.

## Population genetic analysis

To analyze population genetics, we directly downloaded the SNP set of rice accessions from a previously published work[49] and converted the SNP coordinates of IRGSP4 to MSU RGAP 7 using BWA, which was used to process the local alignment. The SNPs covering the whole 200-kb chromosomal region upstream and downstream of *HWS1* in the *japonica* and *indica* accessions of *O.sativa* and three types of wild rice were extracted to calculate the nucleotide diversity (π) using VCFtools in sliding windows with a 5-kb window size[73]. The gene body and a total of 1.2-kb of DNA sequence upstream and downstream of *HWS1* were extracted using BEDTools utilities, and the sequences were aligned using MAFFT software[74]. The corresponding variants from 41 rice varieties in this small genomic interval were then identified using custom scripts (simple 'shell' command) to compute genome-wide linkage disequilibrium (LD). We next performed case-control genetic association analysis and calculated the LD to test the correlation by

Pearson's Chi-square test in PLINK version 1.90[75]. The LD block was displayed using LDBlockShow software[76].

We downloaded all corresponding well-assembled genomes of 32 Asian rice varieties and 20 African rice varieties and then mapped them to the *O. sativa* 'Nipponbare' reference genome (IRGSP1.0) (https://rapdb.dna.affrc.go.jp/) using MuMmer (version 4.0) (parameters: NUCmer -l 50 -c 100 -maxmatch)[77]. A delta-filter program with a one-to-one alignment model (-1) was provided to manipulate the filter of the alignment blocks and the resulting SNPs were identified accordingly using show-snps (-ClrT) in the MUMmer toolkit. Admixture among the 71 *O.rufipogon* accessions and another 12 *O. sativa* cultivars from our new de novo genome assemblies were also selected to gather the SNP information.

## ABBA-BABA statistic test
In the analysis of introgression, we estimated the fd statistical values using the python script ABBABABAwindows.py under topology ((X, Y), *O.barthii*, *O.meridionalis*), where X, Y are extracted from Or-III, Or-II, and Or-I, respectively. The parameters were set as "window size: 300-K bp, minimum good sites per window: 100, and minimum proportion of samples genotyped per site: 0.2. We defined a region as being introgressed if fd was in the top 0.5% of the genome-wide distribution.

## Statistics and reproducibility
Statistical analysis and number of biologically independent sample (n) were indicated in the figure legends. Significant differences between two groups were determined with two-sided Student's paired *t* test by using Microsoft Excel 2013 software. Differences between three or more groups were determined by using one-way ANOVA with two-sided Tukey's HSD test with GraphPad Prism 8 software.

## Reporting summary
Further information on research design is available in the Nature Portfolio Reporting Summary linked to this article.

## Data availability
All the relevant data supporting the findings of this work are avaliable in this article and its Supplementary information files. The RNA-seq and ChIP-seq data generated in this study have been deposited at the National Genomics Data Center under accession numbers (ID: CRA014308, CRA014315). Source data are provided with this paper.

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

## Acknowledgements

We thank M. Shi for assistance with the transgenic assay; W. J. Cai, S. N. Yin, Z. P. Zhang, X. Y. Gao, J. Q. Li, and L. N. Xu for technical supports; Y. G. Liu for donation of CRISPR/Cas9 plasmids; L.G. Shang and Q. Qian for providing the rice super pan-genome data; S. G. Li and P. Qin for providing the geographic location of 33 *O. sativa* accessions. This work was supported by the grants from National Natural Science Foundation of China (32388201), the Chinese Academy of Sciences (XDB27010104,

159231KYSB20200008), Laboratory of Lingnan Modern Agriculture Project (NT2021002), CAS-Croucher Funding Scheme for Joint Laboratories.

## Author contributions

H.X.L. and B.H. conceived and supervised the project, and H.X.L., Y.H.X. and B.L. designed the experiments. B.L., Y.H.X. and Y. L. performed most of the experiments. K.Y.Y., J.X.S., W.W.Y., N.Q.D., Y.K., Y.B.Y., H.Y.Z., H.X.Y., Z.Q.L., Y.Z., Q.Z., D.G., S.Q.G., J.J.L., X.R.M., Y.J.C. and H.X.L. performed some of the experiments. Y.H.X., B.L. and H.X.L. analyzed data and wrote the manuscript.

## Competing interests

The authors declare no competing interests.
