## [Peer Review File · Nature Communications]

REVIEWER COMMENTS

Reviewer #1 (Remarks to the Author):

This paper is a comprehensive genetic analysis of a pair of paralogues that appear to be partially responsible for reproductive isolation between *O. sativa* and *O. glaberrima*. The two genes are termed RIS1 and RIS2, and these are paralogous genes in which loss-of-function confers complete interspecific incompatibility between Asian and African rice. These genes encode putative Esa1-associated factor 6 (EAF6) proteins which act as a subunit of the histone H4 acetyltransferase complex regulating histone modification and thus transcriptional activation. The analysis shows defective tapetum development leading to abnormal seeds and pollen in F2 hybrid offspring due to the RIS1/2-mediated mis-regulation of lipid synthesis and vitamin metabolism.

This is a very good manuscript in many ways. First it shows convincingly that RIS1 and RIS2 interact epistatically to lead to sterility. It then does a good job in providing some functional characterization of the genes that provide a mechanistic basis for the reproductive isolation. A few comments here – it would be good at the beginning of the results if they would explicitly define their genotypes (G1G1, G2G2, W2G2, etc.) as it became hard to follow without them laying out that G meant *glaberrima* (?) and 1 meant the gene (?). So which genotypes are sterile? Second, I would turn away from the term “non-Mendelian inheritance” which may suggest some sort of strange epigenetic phenomena – I would just say the straightforward “segregation distortion.”

The area that was a little less compelling was the evolutionary analysis. I have a few comments:

1. The way it was written I felt was confusing as you tried to wade through Class I and II and III and the species and the genes, etc. I think the entire section should be worked on to make the evolutionary story clearer. Also, Class designation seems to be based on genotypes while they were discussing it in terms of genes – I would eliminate this and simply discuss the evolution of genes/alleles, and then later look at how they come together as genotypes. Remember – it is genes and alleles that are inherited, not genotypes. This would make the discussion more straightforward.

2. The K_s values ($K_s > 0.5$) seemed very high. It would have been close to the saturation levels, and therefore make some of their inferences problematic. They certainly are very high for alleles! If the genes are that diverged, they should also try to look at the phylogeny based on K_a values which would be further from saturation. By the way, Figure 4c to me was confusing as to what it really showed.

3. The case for introgression – they state that “By looking up their allelic information in each accession of Class III, we found universal evidence of introgression from African rice in most of the selected *O. rufipogon*, especially in the Or-III type *O. rufipogon*.” It is not clear at all what this “universal evidence” is. There is also the possibility that this may be lineage sorting. There are strong statistical tests for introgression, and they should use these in their analyses.

4. If RIS1 and RIS2 alleles are really important for speciation, then they should provide estimates of the frequency of the interacting sterile alleles in the two species. This is crucial – to be a major factor in speciation, the alleles have to be at or near-fixation.

5. Finally, I don't think there is good evidence for the involvement of transposons. Just because there is a lot of them around one of the duplicates is interesting but not really string evidence.

Reviewer #2 (Remarks to the Author):

Reviewers' comments:

This study uncovered an intriguing mechanism that the reciprocal gene loss of duplicated genes encoding a subunit of the histone H4 acetyltransferase complex confers hybrid weakness between Asian and African cultivated rice. As shown by the work, the data obtained from genetic and molecular analyses are sound and convincing, revealing that dysfunction of the pair genes of histone H4 acetyltransferase locus cause hybrid weakness resulting from mis-regulation of lipid synthesis and vitamin metabolism. The subsequent sequence and divergence analysis describes an evolutionary picture with respect to the origin of such reproductive barrier. In addition, the neutral rice germplasm containing two duplicated functional EAF6 genes is useful in interspecific hybrid breeding. The study is solid and most experiments are well designed, well carried out and clearly presented. Several suggestions are proposed as follows.

Major suggestions

1. Unfortunately, the causal gene EAF6 has been cloned by Nori Kurata's group (Kubo et al., 2022), which makes the study less novelty. Although this manuscript is more complete and detailed, the same paired genes and the similar evolutionary picture have been found in indica-japonica hybrids by Kubo et al.

2. In this manuscript, the authors showed that loss-function of the paired genes of histone acetyltransferase leads to segregation distortion and hybrid weakness. The genetic data on that the gene loss of duplicated genes EAF6 causes hybrid weakness is solid and convincing. However, the evidences for segregation distortion are unclear. The observed segregation of RIS1 locus in the self-pollinated progeny of NIL-RIS1/RIS2 was 226: 570: 322 (W1W1: W1G1: G1G1) without regard to the genotype of RIS2 locus, indicating that the sterility effect caused by RIS1 or RIS2 locus is very limit in the hybrid F1

plants. The segregation distortion of RIS1 and RIS2 locus indeed existed only in these plants with the homozygous ris2CG14 genotype and ris1WYJ7, respectively, suggesting that RIS1 hybrids with homozygous ris2CG14 and RIS2 hybrids with homozygous ris1WYJ7 may display male or female semi-sterility. But the authors clarified that these hybrids are fertile. This should be explained.

3. The expression pattern of the RIS1 and RIS2 in pollen grain might be provided to facilitate understanding their function on gametophytic development. In addition, the data showed that ris1WYJ7 is a null allele due to that the transcripts were undetectable. The promoter activity of ris1WYJ7 might be provided to confirm that the promoter mutation lead to failure of expression.

4. The authors should change the gene locus names from RIS1 (Reproductive Isolation 1) and RIS2 (Reproductive Isolation 2) to appropriate names, because these names with "1", "2" will produce misreading that this type of reproductive isolation genes are the firstly discovered case. In fact, there are a number of postzygotic reproductive isolation loci, including the similar type of duplicated loci with reciprocal gene loss have been reported (refs 13, 14, 16, 48).

Minor suggestions

1. Some details should be provided for the description of RNA-seq and ChIP-seq analysis. I could not find the information of biological replicates, the number of individuals, and other related information in the method section and related figure legend.

2. Line 114, for the consistence with the unit of 32.5-kb, 1.35-cM should be changed into ## kb.

3. Given that the same genes EAF6 of RIS1/RIS2, and other duplicated male sterility gene pairs (refs 13, 14, 16, 48) have been previously cloned and reported, I strongly recommend to elaborate a better discussion in the context of the previous literature, indicating what are the new findings in this study in comparing with the previous related reports.

Reviewer #3 (Remarks to the Author):

In this manuscript, the authors reported a couple of duplicated paralogs, RIS1 and RIS2, responsible for hybrid weakness between Asian and African rice. Although the map-based cloning of the two identical genes (HWE1 and HWE2) has been published recently (Kubo et al., 2022, Front. Plant Sci. 13:866404), the authors focused on the gene function in hybrid incompatibility and evolutionary analysis and obtained some interesting new data. My concerns about this manuscript are listed below.

1. Eight SNPs were identified in the association analysis of the expression level and nucleotide variations of RIS1, with SNP1_7385301 (T-to-C substitution) showing the strongest signal. Although the sterility and

weakness phenotypes in the hybrid combinations between SG178 and seven cultivars coincided with the claim that SNP1_7385301 was the causative variant of RIS1 expression (Fig. 5), no direct evidence about the function of this SNP was presented. As further classification of different RIS groups and evolutionary analysis is somehow based on this SNP, I suppose that the authors need to offer more experimental evidence to support the function of the SNP (and/or other SNPs), such as using complementation and/or base editing approaches to verify the assumption.

2. Another important claim in the manuscript is the introgression of the RIS1 locus from African rice to Asian rice based on the highly consistent sequence similarity at the RIS1 locus between Class-III Asian rice (ris1C type) and African rice (Extended data Fig. 8a). If it is the case, it seems to me that there should be at least some RIS1T haplotypes in Class-I and Class-II Asian rice (especially *O. rufipogon*) with higher sequence similarity to African rice than ris1C haplotypes in Class-III Asian rice, because African rice belongs to the Class-II group (RIS1T type). In addition, the event of this T-to-C substitution in RIS1 is somehow missing in the evolutionary study. From the Extended Data Fig. 8b, the T-to-C substitution seems to have happened independently, as there are many Class-III *O. rufipogon* without introgression signal at the RIS1 locus. I noticed that there was always a cyclinA1 gene next to RIS1 or RIS2. Is it possible that introgression analysis could also be done based on the localized sequence comparison of the cyclinA1-RIS genomic block in addition to the chromosome-level introgression footprint analysis or single gene phylogenetic analysis of RIS?

3. In the proposed functional model (Extended Data Fig. 9), the author focused on the sporophytes (2n) with the ris1C/ris2Null allelic combination which show reduced fitness and complete sterility due to improperly established chromatin status. However, RIS1/2 is also supposed to be functional in the gamete phase (n), which is not clearly presented in the proposed model. Cross combinations of RIS1/ris1 ris2/ris2, ris1/ris1 RIS2/ris2, and RIS1/ris1 RIS2/ris2 with WYJ7 or SG178 would be able to clarify the impact of ris1ris2 on eliminated transmission of specific gametes and explain the 7:8:1 segregation ratio in Fig. 1a, providing more support for the RIS1/2-mediated reproductive isolation system.

4. Double recessive gene combination, ris1ris2, causes weak growth phenotype and complete sterility in F2 generation, which is a typical phenomenon of hybrid weakness. Plant defense systems such as autoimmune responses are common causes of hybrid weakness; therefore, I am wondering whether genes involved in autoimmune responses are also enriched in the RNA-seq or ChIP-seq analyses.

5. In Line 445, "The overlap between H4Ac-enriched genes and transcriptional DEGs only occupied a small proportion, suggesting the EAF6 may also play a part in acetylating nucleosomal H2A or H3 as well for transcriptional activation (Fig. 3e)." The authors are suggested to do a simple western blot experiment (similar to Fig. 3c) to verify this assumption. In addition, NIL-RIS1 and/or NIL-RIS1RIS2 can also be included in the western blot analysis.

6. The peak of H4Ac enrichment is in the downstream of TSS (Extended Data Fig. 6e), but in Extended Data Fig. 6f, the majority (88.98%) of the selected H4Ac-enriched peaks are localized at the promoter region.

7. In Fig. 4b, the gene tree didn't coincide with the species tree, especially for RIS2. The RIS1 sequence was from japonica (Class-III group); Will the position of RIS1 be different if the RIS1 sequences of Class-I and/or Class-II groups are used in phylogenetic tree construction?

8. In Fig. 4e, the authors mentioned that japonica population might suffer from the bottleneck effect. If Class-III *O. rufipogon* is used in the same analysis, will a strong positive selection signal also be found at the RIS1 locus? If yes, then why is a nonfunctional *ris1* in Class-III *O. rufipogon* positively selected?

9. Specify the "use or lose" strategy (Line 418) as there are a large proportion of accessions (30.83%) with both RIS genes functional in Class-I group (Fig. 5j). Cite original references if applicable.

10. In Fig. 5j, all indica has the RIS1T allele, while 12/13 japonica has the *ris1C* allele. Is this SNP involved in indica-japonica differentiation?

Other minor points:

1. In Supplementary Table 1, the sum of the expected number doesn't equal the sum of the observed number. In addition, how do you calculate the Chi-square values for each genotype in a segregating population?

2. In Supplementary Table 2, there are many accessions with more than one copy of RIS1 or RIS2, but the numbers of the Ks value do not completely match the gene copy numbers, such as W1559, W1666, W2051, ...

3. In Line 150, the word "introgressed" gave me the impression that the null allele of 310 existed in another variety and was crossed into the WYJ7 background, while it was directly created by gene editing instead.

4. In Line 493, the website "www.RiceRC.com" can not be opened.

5. In Line 668-681, the authors didn't specify the materials used in protein extraction and western blotting.

6. In Extended Data Fig. 1a, "RIS1WYJ7" should be "RIS1CG14".

7. In Extended Data Fig. 1c, the pollen of CG14 seems to be abnormal. Replace it if you have additional figures.

8. In Extended Data Fig. 3b, is the tree built based on the EAF4 domain or the whole protein sequence?

9. In Extended Data Fig. 4a-b, we can determine that RIS genes are constantly expressed in different materials, but it is hard to determine the actual expression level compared to Ubiquitin, because all the relative expressions were normalized twice. More information could be obtained if the authors only normalize the expression data once.

10. In Extended Data Fig. 6h, there are two "Membrane-bounded organelle" items.

Point-by-point Response to Reviewers

Dear Reviewers,

We would like to thank all of you for your constructive comments and valuable suggestions on improving our manuscript. We have performed several additional experiments and made the necessary modifications carefully as recommended. Hopefully, we are able to eliminate your concerns and fully address the questions you raised based on the corresponding new data we provide here. The major changes are all highlighted in blue in the revised version of our manuscript, and will not affect the overall framework and content of the paper. We hope that the revised manuscript is more acceptable and satisfactory. Detailed point-by-point responses to the comments are provided below.

Reviewer #1:

This paper is a comprehensive genetic analysis of a pair of paralogues that appear to be partially responsible for reproductive isolation between *O. sativa* and *O. glaberrima*. The two genes are termed RIS1 and RIS2, and these are paralogous genes in which loss-of-function confers complete interspecific incompatibility between Asian and African rice. These genes encode putative Esa1-associated factor 6 (EAF6) proteins which act as a subunit of the histone H4 acetyltransferase complex regulating histone modification and thus transcriptional activation. The analysis shows defective tapetum development leading to abnormal seeds and pollen in F2 hybrid offspring due to the RIS1/2-mediated mis-regulation of lipid synthesis and vitamin metabolism.

This is a very good manuscript in many ways. First it shows convincingly that RIS1 and RIS2 interact epistatically to lead to sterility. It then does a good job in providing some functional characterization of the genes that provide a mechanistic basis for the reproductive isolation.

A few comments here – it would be good at the beginning of the results if they would explicitly define their genotypes (G1G1, G2G2, W2G2, etc.) as it became hard to follow without them laying out that G meant *glaberrima* (?) and 1 meant the gene (?). So which genotypes are sterile?

Response: Thank you for your suggestion, and we apologize for not having described the genotypes succinctly. “G” and “W” are corresponded to the CG14 (*O.glaberrima*) and WYJ7 (*O.sativa*) genotypes respectively, wherein the number “1” and “2” represent *HWS1* (*Hybrid Weakness and Sterility*; original gene name, *RIS1*) or *HWS2* (original gene name, *RIS2*) locus. Consequently, G1G1 or G2G2 means homozygous CG14-type genotype at the *HWS1* or *HWS2* locus; Similarly, W1W1 or W2W2 means homozygous WYJ7-type genotype at the *HWS1* or *HWS2* locus; and W1G1 or W2G2 means heterozygote at *HWS1* or *HWS2* locus. The offspring possessing the W1W1|G2G2 genotype are sterile accordingly. We have defined the genotypes at the beginning of the

results in the revised manuscript (please see **Lines 85-87; 90-91; 97-98**).

Second, I would turn away from the term “non-Mendelian inheritance” which may suggest some sort of strange epigenetic phenomena – I would just say the straightforward “segregation distortion.”

Response: Thank you for pointing this out. We have changed the phrase “non-Mendelian inheritance” to “segregation distortion” as recommended (please see **Lines 2 and 187**).

The area that was a little less compelling was the evolutionary analysis. I have a few comments:

1. The way it was written I felt was confusing as you tried to wade through Class I and II and III and the species and the genes, etc. I think the entire section should be worked on to make the evolutionary story clearer. Also, Class designation seems to be based on genotypes while they were discussing it in terms of genes – I would eliminate this and simply discuss the evolution of genes/alleles, and then later look at how they come together as genotypes. Remember – it is genes and alleles that are inherited, not genotypes. This would make the discussion more straightforward.

Response: We appreciate your considerate suggestions, and have re-combed the whole evolution part in an accurate and clear description. We carefully went over the entire manuscript and have avoided to mention Class I-III ahead of time, and revised these corresponding sentences (please see **Lines 324; 327; 351; 368; 372**). Besides, to explain the shared genetic variation between Asian and African rice, we have corrected and provided the newest evidence about the proposed incompatible sorting lineage event leading to phylogenetic conflicts at the *HWSI* locus instead of introgression (please see the newly added **Fig. 4c-e; Lines 329-341** and the newly added **Extended Data Fig. 10b-d; Lines 342-356**).

2. The K_s values ($K_s > 0.5$) seemed very high. It would have been close to the saturation levels, and therefore make some of their inferences problematic. They certainly are very high for alleles! If the gens are that diverged, they should also try to look at the phylogeny based on K_a values which would be further from saturation. By the way, Figure 4c to me was confusing as to what it really showed.

Response: Thank you for your insightful comments. We speculated that the high K_s values attributed to the various calculation objects considering that K_s values of *HWSI* were relative to syntenic orthologs in *Leersia perrieri*, a species that is widely accepted as the ancestor of rice. Alternatively, we selected *O. meridionalis* (an ancient wild rice species with the AA genome) (please see **Lines 362-364**) as a replacement and recalculated the K_s values. The updated K_s results have been used to replace the original **Fig. 4c** (please see the newly updated **Fig. 4f** and **Extended Data Fig. 10e**;

Lines 362-370).

As a footnote, the calculated Ks values in **Fig. 4f** and **Extended Data Fig. 10e** were presented to illustrate the duplication event of *HWS1/2* genes and speciation referring to the previous studies (Navarro and Barton, 2003; Stein et al., 2018). We calculated the Ks values of gene pairs between *HWS1* and *HWS2* from selected varieties relative to the *HWS1* of *O. meridionalis* and the Ks values were negatively correlated with kinship with its ancestral copy. The results showed that respective *HWS1* of *O. rufipogon* and *O. barthii* were diverged from *O. meridionalis* separately (0.028 and 0.053) (please see the revised **Fig. 4f; Lines 362-367**), while *HWS2* was subsequently replicated from *HWS1* by duplication in *O. rufipogon* (0.035) (please see the revised **Fig. 4f; Lines 367-368**). Meanwhile, Class III type of *O. rufipogon* (*hws1^C*) varieties encountered the ILS event leading to the similar sequence polymorphism between the inherited *japonica* population and African rice at *HWS1* locus (0.049) (please see the newly added **Extended Data Fig. 10e; Lines 368-369**).

References:

Navarro, A., and Barton, N.H. (2003). Chromosomal Speciation and Molecular Divergence--Accelerated Evolution in Rearranged Chromosomes. *Science* 300, 321-324.

Stein, J.C., Yu, Y., Copetti, D., Zwickl, D.J., Zhang, L., Zhang, C., Chougule, K., Gao, D., Iwata, A., Goicoechea, J.L., et al. (2018). Genomes of 13 domesticated and wild rice relatives highlight genetic conservation, turnover and innovation across the genus *Oryza*. *Nature Genetics* 50, 285-296.

3. The case for introgression – they state that “By looking up their allelic information in each accession of Class III, we found universal evidence of introgression from African rice in most of the selected *O. rufipogon*, especially in the Or-III type *O. rufipogon*.” It is not clear at all what this “universal evidence” is. There is also the possibility that this may be lineage sorting. There are strong statistical tests for introgression, and they should use these in their analyses.

Response: Thank you for your comments, and we feel appreciated for the constructive and insightful suggestions. Both incomplete lineage sorting (ILS) and introgression can result in phylogenetic discrepancies between gene tree and species tree, and researchers can easily be confused and misguided because of similar patterns of shared genetic diversity (Feng et al., 2022). Based on your advice, we have performed the ABBA-BABA statistical test for the introgression analysis. No obvious introgression signal was observed in the window where the *HWS1* gene was located (please see newly added **Extended Data Fig. 10d; Lines 351-356**). Instead, we compared the time of speciation with *HWS1* gene divergence time, and the results showed that incomplete lineage sorting (ILS) may be the major contributor to the inconsistency between gene tree and species tree. We have corrected and revised this part by removing the original **Extended Data Fig. 8b** and **Extended Data Fig. 8c** (please see newly added **Fig. 4c-e** and **Extended Data Fig. 10b, c; Lines 329-349**).

Reference:

Feng, S., Bai, M., Rivas-González, I., Li, C., Liu, S., Tong, Y., Yang, H., Chen, G., Xie, D., Sears, K.E., et al. (2022). Incomplete lineage sorting and phenotypic evolution in marsupials. *Cell* 185, 1646-1660.e1618.

4. If RIS1 and RIS2 alleles are really important for speciation, then they should provide estimates of the frequency of the interacting sterile alleles in the two species. This is crucial – to be a major factor in speciation, the alleles have to be at or near-fixation.

Response: Thank you for this valuable suggestion. According to our evolution analysis, all the detected African rice, including 12 *O.glaberrima* and 8 *O.barthii* that belong to the Class II type, possess 100% *HWS1^T* type allele and lack *HWS2* (please see **Fig. 5k**). We also investigated the genotypes of extra 103 *O.glaberrima* (Huang et al., 2015), and found that *HWS1^T* allele accounts for 85.4% and no *hws1^C* allele is existing (please see the newly added **Supplementary Table 5**). The *HWS1^T* allele did not reach 100%, most likely because of the 14.6% missing data. For the *japonica* group, the frequency of the *hws1^C* allele is nearly 96.8% and *HWS2* retains based on the public rice database (MBKbase, Rice) (please see the newly added **Supplementary Table 5**). The above results confirmed that the sterile allele *hws1^C* and the functional allele *HWS2* are or near fixed in the *japonica* group and the functional allele *HWS1^T* is or near fixed in the African rice group (please see the newly added **Supplementary Table 5; Lines 453-455**).

Reference:

Huang, X., Zhao, Q., and Han, B. (2015). Comparative Population Genomics Reveals Strong Divergence and Infrequent Introgression between Asian and African Rice. *Molecular Plant* 8, 958-960.

5. Finally, I don't think there is good evidence for the involvement of transposons. Just because there is a lot of them around one of the duplicates is interesting but not really string evidence.

Response: Thank you for your comments. From our newly added synteny analysis, *HWS1* came out of nowhere on the chromosome segment (please see the newly added **Extended Data Fig. 8c; Lines 309-314**), while *HWS2* has no obvious syntenic relationships between the analyzed species (please see the newly added **Extended Data Fig. 8d; Lines 314-316**). It is reasonable to speculate that the *HWS1/2* gene pair was generated by transposon-mediated gene replication when considering that transposable elements (TEs) generally cause no synteny (Feng et al., 2022). Additionally, We investigated the distribution of all DNA transposons and LTR retrotransposons in the upstream and downstream 10kb region of *HWS1/2* gene of 32 rice varieties and found that only DNA transposon (*Os1417*) existed in the upstream and downstream regions of each gene, indicating the positive effect of this transposon to the gene duplication (please see **the following Fig. 1**). This result is accordance with the previously reported

pattern of transposon-mediated gene replication (Catoni et al., 2019; Jiang et al., 2004). We agree that we cannot offer direct evidence at present which requires further studies, but the transposon annotation and collinearity analysis strongly suggested that the transposon *Os1417* may have a positive effect on *HWS* gene duplication and horizontal gene transfer. Thus, we summarized and described this part of the conclusion as: “*OsEAF6* was probably generated by DNA transposon-mediated gene replication” (please see **Line 321**) in the manuscript. We sincerely hope that you can understand this limitation.

Fig. 1 (for reviewers only) Transposons distribution upstream and downstream in 10-kb region around *HWS2* locus in 32 rice varieties. The annotated transposons are listed in the right and the *Os1417* transposon is colored in red. *HWS2* locus of 32 rice varieties is listed at the bottom. The blue and grey squares represent the presence and absence of transposons while the red and cyan squares represent upstream and downstream of the *HWS2* locus in each rice variety.

Reference:

Jiang, N., Bao, Z., Zhang, X., Eddy, S.R., and Wessler, S.R. (2004). Pack-MULE transposable elements mediate gene evolution in plants. *Nature* 431, 569-573.

Catoni, M., Jonesman, T., Cerruti, E., and Paszkowski, J. (2019). Mobilization of Pack-CACTA transposons in Arabidopsis suggests the mechanism of gene shuffling. *Nucleic Acids Research* 47, 1311-1320.

Feng, S., Bai, M., Rivas-González, I., Li, C., Liu, S., Tong, Y., Yang, H., Chen, G., Xie, D., Sears, K.E., et al. (2022). Incomplete lineage sorting and phenotypic evolution in marsupials. *Cell* 185, 1646-1660.e1618.

Reviewer #2:

This study uncovered an intriguing mechanism that the reciprocal gene loss of duplicated genes encoding a subunit of the histone H4 acetyltransferase complex confers hybrid weakness between Asian and African cultivated rice. As shown by the work, the data obtained from genetic and molecular analyses are sound and convincing, revealing that dysfunction of the pair genes of histone H4 acetyltransferase locus cause hybrid weakness resulting from mis-regulation of lipid synthesis and vitamin metabolism. The subsequent sequence and divergence analysis describes an evolutionary picture with respect to the origin of such reproductive barrier. In addition, the neutral rice germplasm containing two duplicated functional *EAF6* genes is useful in interspecific hybrid breeding. The study is solid and most experiments are well designed, well carried out and clearly presented. Several suggestions are proposed as follows.

Major suggestions

1. Unfortunately, the causal gene *EAF6* has been cloned by Nori Kurata's group (Kubo et al., 2022), which makes the study less novelty. Although this manuscript is more complete and detailed, the same paired genes and the similar evolutionary picture have been found in indica-japonica hybrids by Kubo et al.

Response: Thanks for your comment. Despite the fact that the *EAF6* genes have been cloned elsewhere, our work has expanded the applicability of the *EAF6*-mediated hybrid breakdown model to a wider base of inter-subspecific incompatibility. We not only systematically investigated the copy number variation of the *EAF6* gene in Asian and African rice (please see **Extended Data Fig. 9a, b** and **Supplementary Table 2**), but also comprehensively traced the pedigree evolution of *EAF6* in *Oryza* genus (please see the revised **Fig. 6; Lines 417-440**), which are conducive to parent selection in rice breeding-cross performance. Worthy of note in our study is the reduced sequence polymorphism at the *HWSI* locus caused by bottleneck and founder effect in the *japonica* population and the proposed incomplete lineage sorting (ILS) event explaining the shared genetic variation of *HWSI* between Asian and African rice. By evaluating the agronomic traits of *EAF6*-overexpressing lines (please see **Extended Data Fig. 13**), an alternative strategy was proposed to restore seed fertility by altering *EAF6*-DNA copies other than the usual gene editing. In addition, we decipher the generative mechanism of hybrid breakdown caused by the *HWS* gene pair at the epigenetic level and show that the biotin metabolism and fatty acid may affect gametophyte development due to transcription disorder. Undoubtedly, these findings will certainly deepen our understanding of genetic interaction networks involving hybrid breakdown and expand our knowledge of the origin and evolutionary process of rice during speciation, which are innovation spots in our study. We made an in-depth discussion to compare our work with previous studies (please see **Lines 458-471**).

2. In this manuscript, the authors showed that loss-function of the paired genes of histone acetyltransferase leads to segregation distortion and hybrid weakness. The genetic data on that the gene loss of duplicated genes *EAF6* causes hybrid weakness is

solid and convincing. However, the evidences for segregation distortion are unclear. The observed segregation of RIS1 locus in the self-pollinated progeny of NIL-RIS1/RIS2 was 226: 570: 322 (W1W1: W1G1: G1G1) without regard to the genotype of RIS2 locus, indicating that the sterility effect caused by RIS1 or RIS2 locus is very limit in the hybrid F1 plants. The segregation distortion of RIS1 and RIS2 locus indeed existed only in these plants with the homozygous $ris2^{CG14}$ genotype and $ris1^{WYJ7}$, respectively, suggesting that RIS1 hybrids with homozygous $ris2^{CG14}$ and RIS2 hybrids with homozygous $ris1^{WYJ7}$ may display male or female semi-sterility. But the authors clarified that these hybrids are fertile. This should be explained.

Response: Thank you for raising this question. We have redone the segregating analysis and investigated the pollen and spikelet fertility of each *HWS1* (*Hybrid Weakness and Sterility 1*; original gene name, *RIS1*) and *HWS2* (original gene name, *RIS2*) hybrid. Neither heterozygous NIL-*HWS1*^{CG14}/*hws2*^{CG14} (genotype: W1G1|G2G2) nor NIL-*hws1*^{WYJ7}/*HWS2*^{WYJ7} (genotype: W1W1|W2G2) plants display pollen or spikelet semi-sterility (please see **the following Fig. 1**). Still, the segregation ratio in both *HWS1* (W1W1: W1G1: G1G1=34: 220: 136) and *HWS2* (W2W2: W2G2: G2G2=159: 192: 2) self-pollinated F₂ segregating population do not conform with 1:2:1 standard (please see the following **Table 1** and **2**). It is speculated that the induced deviation at *HWS1/2* loci was affected by the transposons nearby and centromere there (please see **Extended Data Fig. 9** and **Extended Data Fig. 1f**) and might occur at the meiosis stage due to chromosomal rearrangement, non-homologous recombination, and gene conversion (Fu et al., 2020). A more severe segregation distortion at the *HWS2* locus compared to that of *HWS1* may support our hypothesis considering that the *HWS2* locus is adjacent to the centromere. Additionally, a biased transmission ratio determined by killer and protector is not employed in this case since EAF6 might not be toxic in gametophytes and be of sporophytic nature, and the non-viable pollen grains in NIL-*hws1*^{WYJ7}/*hws2*^{CG14} (genotype: W1W1|G2G2) plants might be due to the abnormally developed tapetum because of maternal deficiency (please see **Fig. 2j-l, o-p1**).

Fig. 1 (for reviewers only) Phenotypes of the hybrid NIL-*HWS1*^{CG14}/*hws2*^{CG14} and NIL-

hws1^{WYJ7}/HWS2^{WYJ7} plants. Mature plant architecture (upper panels), panicle morphology (middle panels) of hybrid NIL-*HWS1^{CG14}/hws2^{CG14}* and NIL-*hws1^{WYJ7}/HWS2^{WYJ7}*. I₂-KI stained pollen grains indicating fertility are shown in the top right corners. Scale bars = 15 cm/10 cm/100 μm in the upper/middle/top right corner panels, respectively.

Table 1. (for reviewers only) Chi-square test of observed ratio of genotypes in the progenies of the NIL-*HWS1^{CG14}/hws2^{CG14}*

Genotype		Expect Ratio	NO.of individuals		χ^2	Phenotype
HWS1 _M1	HWS2 _K1		Expected	Observe		
W1W1	G2G2	1	97.5	34	41.35641	S ●
W1G1	G2G2	2	195	220	3.2051282	F ●
G1G1	G2G2	1	97.5	136	15.202564	F ●
				n=3		
				df=2		
			$\chi^2_{0.05}(2)=5.991$ $\chi^2_{0.01}(2)=9.21$			

Table 2. (for reviewers only) Chi-square test of observed ratio of genotypes in the progenies of the NIL-*hws1^{WYJ7}/HWS2^{WYJ7}*

Genotype		Expect Ratio	NO.of individuals		χ^2	Phenotype
HWS1 _M1	HWS2 _K1		Expected	Observe		
W1W1	W2W2	1	88.25	159	56.720255	F ●
W1W1	W2G2	2	176.5	192	1.3611898	F ●
W1W1	G2G2	1	88.25	2	84.295326	S ●
				n=3		
				df=2		
			$\chi^2_{0.05}(2)=5.991$ $\chi^2_{0.01}(2)=9.21$			

References

Fu, Q., Meng, X., Luan, S., Chen, B., Cao, J., Li, X., and Kong, J. (2020). Segregation distortion: high genetic load suggested by a Chinese shrimp family under high-intensity selection. *Scientific Reports* 10, 21820.

3. The expression pattern of the RIS1 and RIS2 in pollen grain might be provided to facilitate understanding their function on gametophytic development. In addition, the data showed that *ris1^{WYJ7}* is a null allele due to that the transcripts were undetectable. The promoter activity of *ris1^{WYJ7}* might be provided to confirm that the promoter mutation lead to failure of expression.

Response: Thank you for your constructive comments. We have performed the RNA *in situ* hybridization assays of NIL-*hws1^{WYJ7}/hws2^{CG14}* and NIL-*HWS1^{CG14}/hws2^{CG14}* to precisely determine the expression pattern of *HWS1* in developing anthers. The results showed that *HWS1^{CG14}* expression signals specifically localized in meiotic cells while *hws1^{WYJ7}* RNAs were completely undetectable (please see the newly added **Extended**

Data Fig. 5; Lines 183-185), which is coincided with the data of qRT-PCR assay (please see **Extended Data Fig. 4a**), and further supports the roles of *HWS1/2* in microspores. To confirm that the promoter T/C variant is associated with the expression level of *HWS1*, we generated a range of promoter mutation constructs, each fused with the *LUC* (*luciferase*) reporter gene and introduced into rice protoplasts to monitor promoter activity. The quantitation of firefly luciferase expression showed that the *HWS1*^{CG14}_{pro}: Luc activity was dramatically reduced after the point mutation (“T” to “C”) in SNP1_7385301 site. Instead, the “C” to “T” replacement in *hws1*^{WYJ7} rescued the expression failure of *HWS1* in WYJ7, indicating that this core mutation may exert great impact on the level of promoter activity (please see the newly added **Fig. 5b; Lines 394-401**).

4. The authors should change the gene locus names from RIS1 (Reproductive Isolation 1) and RIS2 (Reproductive Isolation 2) to appropriate names, because these names with “1”, “2” will produce misreading that this type of reproductive isolation genes are the firstly discovered case. In fact, there are a number of postzygotic reproductive isolation loci, including the similar type of duplicated loci with reciprocal gene loss have been reported (refs 13 , 14, 16, 48).

Response: Thank you for your thoughtful suggestion. We have renamed these two loci from *RIS1/2* to *HWS1/2* (*Hybrid Weakness and Sterility 1/2*) and revised this throughout the entire manuscript based on your advice.

Minor suggestions

1. Some details should be provided for the description of RNA-seq and ChIP-seq analysis. I could not find the information of biological replicates, the number of individuals, and other related information in the method section and related figure legend.

Response: We apologize for our negligence in not giving the specific description with adequate information regarding to RNA-seq and ChIP-seq assay. We have added the detailed information accordingly in the revised version of our manuscript (please see **Lines 686-688; 710-711**). We could not get enough NIL-*hws1*^{WYJ7}/*hws2*^{CG14} plants because of the severe segregation distortion (please see **Fig. 1a** and **Supplementary Table 1**), so we designed the ChIP-seq experiment with 3 individual plants but without biological repeat. Additionally, we generated the ChIP-qPCR experiments with three biological repeats (please see **Fig. 3i**) to further verify the ChIP-seq results. We sincerely hope you can understand, many thanks.

2. Line 114, for the consistence with the unit of 32.5-kb, 1.35-cM should be changed into ## kb.

Response: Thank you for your considerate suggestion. We have adjusted the unit of

genomic interval size to the physical distance as suggested (please see the revised **Extended Data Fig. 1f; Lines 117; 561**).

3. Given that the same genes EAF6 of RIS1/RIS2, and other duplicated male sterility gene pairs (refs 13 , 14, 16, 48) have been previously cloned and reported, I strongly recommend to elaborate a better discussion in the context of the previous literature, indicating what are the new findings in this study in comparing with the previous related reports.

Response: Thank you for your thoughtful suggestion. According to your suggestion, we have rewritten this part to fully discuss the new findings and highlights in our work in comparison with the previous studies (please see **Lines 458-471**).

Reviewer #3:

In this manuscript, the authors reported a couple of duplicated paralogs, RIS1 and RIS2, responsible for hybrid weakness between Asian and African rice. Although the map-based cloning of the two identical genes (HWE1 and HWE2) has been published recently (Kubo et al., 2022, Front. Plant Sci. 13:866404), the authors focused on the gene function in hybrid incompatibility and evolutionary analysis and obtained some interesting new data. My concerns about this manuscript are listed below.

1. Eight SNPs were identified in the association analysis of the expression level and nucleotide variations of RIS1, with SNP1_7385301 (T-to-C substitution) showing the strongest signal. Although the sterility and weakness phenotypes in the hybrid combinations between SG178 and seven cultivars coincided with the claim that SNP1_7385301 was the causative variant of RIS1 expression (Fig. 5), no direct evidence about the function of this SNP was presented. As further classification of different RIS groups and evolutionary analysis is somehow based on this SNP, I suppose that the authors need to offer more experimental evidence to support the function of the SNP (and/or other SNPs), such as using complementation and/or base editing approaches to verify the assumption.

Response: Thank you for your constructive comments. We agree with your opinion that the creation of *HWS1* (*Hybrid Weakness and Sterility 1*; original gene name, *RIS1*) promoter variants would further support the functional role of these SNPs genetically, and we are definitely going to edit the promoter region and generate heritable precise editing at these sites as validation. However, this would take a long time to obtain stable transgenic lines to clarify this point and it is not realistic up to now. Alternatively, to confirm that the promoter C/T variant is associated with the expression level of *HWS1*, we generated a range of promoter mutation constructs, each fused with *LUC* (*luciferase*) reporter gene and introduced into rice protoplasts to monitor promoter activity. The quantitation of firefly luciferase expression showed that the *HWS1*^{CG14}pro: Luc activity was dramatically reduced after the point mutation in the SNP1_7385301 site. Instead, the “C” to “T” replacement in *hws1*^{WYJ7} rescued the expression failure of *HWS1* in WYJ7, indicating that this core mutation may exert great impact on the level of promoter activity (please see the newly added **Fig. 5b; Lines 394-401**). We sincerely hope that you can understand given its overlong experimental period. Special thanks for your insightful comments.

2. Another important claim in the manuscript is the introgression of the RIS1 locus from African rice to Asian rice based on the highly consistent sequence similarity at the RIS1 locus between Class-III Asian rice (*ris1*^C type) and African rice (Extended data Fig. 8a). If it is the case, it seems to me that there should be at least some *RIS1*^T haplotypes in Class-I and Class-II Asian rice (especially *O. rufipogon*) with higher sequence similarity to African rice than *ris1*^C haplotypes in Class-III Asian rice, because African rice belongs to the Class-II group (*RIS1*^T type). In addition, the event of this T-to-C substitution in *RIS1* is somehow missing in the evolutionary study. From the Extended

Data Fig. 8b, the T-to-C substitution seems to have happened independently, as there are many Class-III *O. rufipogon* without introgression signal at the RIS1 locus. I noticed that there was always a cyclinA1 gene next to RIS1 or RIS2. Is it possible that introgression analysis could also be done based on the localized sequence comparison of the cyclinA1-RIS genomic block in addition to the chromosome-level introgression footprint analysis or single gene phylogenetic analysis of RIS?

Response: Thank you for pointing this out. We are sorry for the insufficient evidence concerning the introgression. We performed the additional ABBA-BABA statistical test for the introgression analysis, but no obvious introgression signal was observed, suggesting that the introgression is not the cause of the shared genetic variation at *HWS1* locus between Asian rice and African rice (please see the newly added **Extended Data Fig. 10d; Lines 351-356**). Nonetheless, our updated data showed that incomplete lineage sorting (ILS) may be the major contributor to the inconsistency between gene tree and species tree and explains the sequence similarity between Class-III *O. rufipogon* (*hws1^C*) and *O. barthii*. We corrected and revised this part by removing the original **Extended Data Fig. 8b** and **Extended Data Fig. 8c** (please see newly added **Fig 4. c-e** and **Extended Data Fig. 10b, c; Lines 329-351**), and proposed a more reasonable model in **Fig. 6** and described in the manuscript (please see **Lines 417-440**). In addition, we tend to believe that the nearby linked cyclinA1 gene might be the result of transposon-mediated segment replication.

3. In the proposed functional model (Extended Data Fig. 9), the author focused on the sporophytes (2n) with the *ris1^C/ris2^{Null}* allelic combination which show reduced fitness and complete sterility due to improperly established chromatin status. However, RIS1/2 is also supposed to be functional in the gamete phase (n), which is not clearly presented in the proposed model. Crosse combinations of RIS1/*ris1 ris2/ris2*, *ris1/ris1 RIS2/ris2*, and RIS1/*ris1 RIS2/ris2* with WYJ7 or SG178 would be able to clarify the impact of *ris1ris2* on eliminated transmission of specific gametes and explain the 7:8:1 segregation ratio in Fig. 1a, providing more support for the RIS1/2-mediated reproductive isolation system.

Response: Thank you for your thoughtful suggestion. Although the RNA in situ hybridization experiment showed that *HWS1/2* is highly expressed in pollen grains at various development stages (please see newly added **Extended Data Fig. 5; Lines 183-185**), the hybrid NIL-*HWS1^{CG14}/hws2^{CG14}* (genotype: W1G1|G2G2) and NIL-*hws1^{WYJ7}/HWS2^{WYJ7}* (genotype: W1W1|W2G2) plants were normal and the pollen grains were fertile without the weakness or semi-sterility phenotype (please see **the following Fig. 1**). These results indicated that *HWS1/2*-mediated reproductive isolation system is differed from the “killer-protector system”. We speculated that the non-viable pollen grains in NIL-*hws1^{WYJ7}/hws2^{CG14}* (genotype: W1W1|G2G2) plants might be due to the abnormally developed tapetum because of maternal deficiency (please see **Fig. 2j-l, o-p1**). Additionally, the segregation ratio in both *HWS1* (W1W1: W1G1: G1G1=34: 220: 136) and *HWS2* (W2W2: W2G2: G2G2=159: 192: 2) self-pollinated

F₂ segregating population do not conform with 1:2:1 standard (please see the following **Table 1** and **2**) and we could observe a more severe segregation distortion at *HWS2* locus comparing to that of *HWS1* (please see the following **Table 2**). It is speculated that the induced deviation at *HWS1/2* loci was affected by the transposons nearby and centromere (please see **Extended Data Fig. 9** and **Extended Data Fig. 1f**) and might occur at the meiosis stage due to chromosomal rearrangement, non-homologous recombination, and gene conversion (Fu et al., 2020).

Based on your advice, we modified the functional model in **Extended Data Fig. 11** accordingly (please see the revised **Extended Data Fig. 11**). Also, the obtained cross combination of hybrid NIL-*HWS1/HWS2* (genotype: W1G1|W2G2) with WYJ7 (genotype: W1W1|W2W2) (please see the following **Table 3**) showed the segregation ratio of *HWS1* or *HWS2* alone did not contravene the typical mendelian inheritance, indicating that the segregation distortion may be possibly caused by the non-allelic interaction between *HWS1* and *HWS2* (Kubo and Yoshimura, 2002). We are sorry that we cannot generate and include all the cross combinations you suggested in such a short time, we sincerely hope you can understand, many thanks.

Fig. 1 (for reviewers only) Phenotypes of the hybrid NIL-*HWS1*^{CG14}/*hws2*^{CG14} and NIL-*hws1*^{WYJ7}/*HWS2*^{WYJ7} plants. Mature plant architecture (upper panels), panicle morphology (middle panels) of hybrid NIL-*HWS1*^{CG14}/*hws2*^{CG14} and NIL-*hws1*^{WYJ7}/*HWS2*^{WYJ7}. I₂-KI stained pollen grains indicating fertility are shown in the top right corners. Scale bars = 15 cm/10 cm/100 μm in the upper/middle/top right corner panels, respectively.

Table 1. (for reviewers only) Chi-square test of observed ratio of genotypes in the progenies of the NIL-*HWS1*^{CG14}/*hws2*^{CG14}

Genotype		Expect Ratio	NO.of individuals		χ^2	Phenotype
HWS1 _M1	HWS2 _K1		Expected	Observe		
W1W1	G2G2	1	97.5	34	41.35641	S ●
W1G1	G2G2	2	195	220	3.2051282	F ●
G1G1	G2G2	1	97.5	136	15.202564	F ●
				n=3		
				df=2		
			$\chi^2_{0.05}(2)=5.991$		$\chi^2_{0.01}(2)=9.21$	

Table 2. (for reviewers only) Chi-square test of observed ratio of genotypes in the progenies of the NIL-*hws1*^{WYJ7}/*HWS2*^{WYJ7}

Genotype		Expect Ratio	NO.of individuals		χ^2	Phenotype
HWS1 _M1	HWS2 _K1		Expected	Observe		
W1W1	W2W2	1	88.25	159	56.720255	F ●
W1W1	W2G2	2	176.5	192	1.3611898	F ●
W1W1	G2G2	1	88.25	2	84.295326	S ●
				n=3		
				df=2		
			$\chi^2_{0.05}(2)=5.991$		$\chi^2_{0.01}(2)=9.21$	

Table 3. (for reviewers only) Genotype of F₁ generation in the cross combination of NIL-*HWS1*/*HWS2* × WYJ7

Year	Cross combination	HWS1 locus genotype		W1G1:W1W1 Ratio	χ^2 (1:1) of HWS1 locus	HWS2 locus genotype		W1G1:W1W1 Ratio	χ^2 (1:1) of HWS2
		W1G1	W1W1			W2G2	W2W2		
2019 winter	NIL- HWS1 / HWS2 (genotype: W1G1 W2G2) × WYJ7 (genotype: W1W1 W2W2)	6	7	0.86:1	0.0196	5	8	0.63:1	0.1369
	NIL- HWS1 / HWS2 (genotype: W1G1 W2G2) × WYJ7 (genotype: W1W1 W2W2)	10	4	2.5:1	2.25	7	7	1:1	0
2020 summer	NIL- HWS1 / HWS2 (genotype: W1G1 W2G2) × WYJ7 (genotype: W1W1 W2W2)	3	9	0.33:1	0.4489	6	6	1:1	0
n=3 df=2		$\chi^2_{0.05}(2)=5.99$		$\chi^2_{0.01}(2)=9.21$					

Reference:

Kubo, T., and Yoshimura, A. (2002). Genetic basis of hybrid breakdown in a Japonica/Indica cross of rice, *Oryza sativa* L. *Theoretical and Applied Genetics* 105, 906-911.

Fu, Q., Meng, X., Luan, S., Chen, B., Cao, J., Li, X., and Kong, J. (2020). Segregation distortion: high genetic load suggested by a Chinese shrimp family under high-intensity selection. *Scientific Reports* 10, 21820.

4. Double recessive gene combination, *ris1ris2*, causes weak growth phenotype and

complete sterility in F2 generation, which is a typical phenomenon of hybrid weakness. Plant defense systems such as autoimmune responses are common causes of hybrid weakness; therefore, I am wondering whether genes involved in autoimmune responses are also enriched in the RNA-seq or ChIP-seq analyses.

Response: Thank you for raising this question. Autoimmune responses associated with *NB-LRR* disease-resistance genes or *R* gene are the most common and major causes of hybrid weakness and breakdown, which usually present severe lesions due to programmed cell death (PCD) in the basal nodes (Chen et al., 2014) and leaf (Chae et al., 2014) tissues. However, we did not observe any visible and typical PCD induced in the leaf and basal nodes of sterile plants compared to fertile plants (please see the newly added **Extended Data Fig. 12a-f; Lines 501-506**). In addition, we also did not find any obvious upregulated or downregulated DEGs involving in autoimmune or defense response enriched in the RNA-seq or ChIP-seq data, suggesting that this weak phenotype may not result from autoimmunity (please see **Fig. 3f** and the newly added **Extended Data Fig. 12g; Lines 506-509**).

References

Chae, E., Bomblies, K., Kim, S.-T., Karelina, D., Zaidem, M., Ossowski, S., Martín-Pizarro, C., Laitinen, Roosa A.E., Rowan, Beth A., Tenenboim, H., et al. (2014). Species-wide Genetic Incompatibility Analysis Identifies Immune Genes as Hot Spots of Deleterious Epistasis. *Cell* 159, 1341-1351.

Chen, C., Chen, H., Lin, Y.-S., Shen, J.-B., Shan, J.-X., Qi, P., Shi, M., Zhu, M.-Z., Huang, X.-H., Feng, Q., et al. (2014). A two-locus interaction causes interspecific hybrid weakness in rice. *Nature Communications* 5, 3357.

5. In Line 445, “The overlap between H4Ac-enriched genes and transcriptional DEGs only occupied a small proportion, suggesting the EAF6 may also play a part in acetylating nucleosomal H2A or H3 as well for transcriptional activation (Fig. 3e).” The authors are suggested to do a simple western blot experiment (similar to Fig. 3c) to verify this assumption. In addition, NIL-RIS1 and/or NIL-RIS1RIS2 can also be included in the western blot analysis.

Response: Thank you for your valuable suggestions to improve our manuscript. We have reperformed the western blot experiment to detect histone H2A, H3 and H4 acetylation levels of pairwise *HWS1/2* NILs using multiple antibodies as suggested, including the newly added antibodies (acetyl-H2AK5, acetyl-H2AK9, and acetyl-H3 antibodies) (please see the revised **Fig. 3c**). The quantified protein abundance results show that both histone H2A and H4 acetylation modification were greatly affected by the loss of *HWS1* and *HWS2* (please see **Lines 241-244**).

6. The peak of H4Ac enrichment is in the downstream of TSS (Extended Data Fig. 6e), but in Extended Data Fig. 6f, the majority (88.98%) of the selected H4Ac-enriched peaks are localized at the promoter region.

Response: Thank you for the comment. Considering that promoters are located near to the TSS (transcriptional start site) of genes, it is reasonable that the majority (88.98%) of the selected H4Ac-enriched peaks are localized at the promoter region.

7. In Fig. 4b, the gene tree didn't coincide with the species tree, especially for RIS2. The RIS1 sequence was from japonica (Class-III group); Will the position of RIS1 be different if the RIS1 sequences of Class-I and/or Class-II groups are used in phylogenetic tree construction?

Response: Thank you for your question. We have reconstructed the gene tree including the Class I-III *HWS1* and *HWS2* of *O. rufipogon* and *O. sativa* for analysis (please see the revised Fig. 4b; Lines 322-324). The positions of Class I/II-*HWS1* are different from Class III-*HWS1* and are consistent with the species tree. Our updated evidence suggested that incomplete lineage sorting (ILS) may be the main reason explaining the phylogenetic conflict between gene tree and species tree (please see the newly added Fig. 4c-e and Extended Data Fig. 10b-d; Lines 329-356).

8. In Fig. 4e, the authors mentioned that japonica population might suffer from the bottleneck effect. If Class-III *O. rufipogon* is used in the same analysis, will a strong positive selection signal also be found at the RIS1 locus? If yes, then why is a nonfunctional *ris1* in Class-III *O. rufipogon* positively selected?

Response: Thank you for raising this question. In the adjacent regions of the *HWS1* gene (7.35Mb-7.4Mb), nucleotide diversity (π) of three wild rice populations (Or I-III) was all significantly higher than that of cultivated rice, indicating a strong positive selection occurring in this region (please see Fig. 4h; Lines 373-375). Since the *HWS1* locus showed very low nucleotide diversity in all wild and cultivated rice populations, we could not confirm whether the *HWS1* locus was positively selected simply from Fig. 4h, so we revised this conclusion in the corresponding section (please see Lines 373-375; 378-380). From our newly proposed evolution model, a T-to-C mutation was generated at random on the promoter region of *HWS1* in the heterozygous Class I-type *O. rufipogon* and eventually contributed to the generation of Class III-type *O. rufipogon* with the nonfunctional *hws1^C* (please see the revised Fig. 6b; Lines 426-438). This nonfunctional *HWS1* locus retains in evolution due to the occurrence of functional *HWS2* in Class III-type *O. rufipogon*. Additionally, we speculate that the extremely low π value of the japonica group might be attributed to the founder effect in the population establishment during domestication, and the reduced nucleotide diversity at the *HWS1* locus in all populations indicates *HWS1* is functionally conserved. This inference may be highly plausible considering that the japonica subspecies underwent a severe bottleneck during the domestication of rice according to the previous study (Huang et al., 2012; Zhu et al., 2007).

Reference:

Zhu, Q., Zheng, X., Luo, J., Gaut, B.S., and Ge, S. (2007). Multilocus Analysis of Nucleotide Variation of *Oryza sativa* and Its Wild Relatives: Severe Bottleneck during Domestication of Rice. *Molecular Biology and Evolution* 24, 875-888.

Huang, X., Kurata, N., Wei, X., Wang, Z.-X., Wang, A., Zhao, Q., Zhao, Y., Liu, K., Lu, H., Li, W., et al. (2012). A map of rice genome variation reveals the origin of cultivated rice. *Nature* 490, 497-501.

9. Specify the “use or lose” strategy (Line 418) as there are a large proportion of accessions (30.83%) with both RIS genes functional in Class-I group (Fig. 5j). Cite original references if applicable.

Response: Thank you for pointing this out. Based on our renewed evolution model, *HWS1* is the ancestral *OsEAF6* copy while *HWS2* originated from *HWS1* and was possibly generated by DNA transposon-mediated gene replication. A T-to-C mutation in the promoter region of *O. barthii*-like (OB-like) *HWS1* allele was produced at random and contributed to the occurrence of the nonfunctional OB-like *HWS1*. By the hybridization event between the Class I-type *O. rufipogon* and the OB-like wild rice ancestors, the Class I-III type *O. rufipogon* was subsequently formed in the self-progeny, which eventually constitutes the *O. rufipogon* gene pool (please see the revised **Fig. 6b; Lines 426-438**). We tend to believe that this gene inactivation may have started as a random event and was finally preserved due to the sufficient function of single-copy EAF6. Considering that there lacks conclusive evidence yet to explain this, we decided to avoid to draw the indeliberate conclusion and have deleted the sentence below “**The allele-specific expression of the *OsEAF6* pair indicates a potential feedback mechanism, that is a "use or lose" strategy in response to cryptic mutation during long-term natural selection**” in the original manuscript.

10. In Fig. 5j, all indica has the $RIS1^T$ allele, while 12/13 japonica has the $ris1^C$ allele. Is this SNP involved in indica-japonica differentiation?

Response: Thank you for raising this question. We investigated the allele distribution of *HWS1* and the other previously reported sterility genes (*RHS12*, *HSA1*, *SaM/F*, *Sc*, and *S5*) controlling *indica-japonica* hybrid sterility in 38 rice varieties (13 *japonica* and 25 *indica* varieties included), and found that all of these sterility genes are differentially distributed in the *indica-japonica* population (please see **the following Fig.1**). We speculated that *HWS1* may serve as the speciation gene and be involved in *indica-japonica* differentiation.

Fig. 1 (for reviewers only) Allele differentiation of *HWS1* and the related *indica-japonica* hybrid sterility genes in 38 rice varieties. 38 rice varieties including 13 *japonica* and 25 *indica* are listed in the right panel. *HWS1* and the reported *indica-japonica* hybrid sterility genes are listed in the bottom. Grey and purple boxes indicate the *japonica* and *indica* allele, respectively.

Other minor points:

1. In Supplementary Table 1, the sum of the expected number doesn't equal the sum of the observed number. In addition, how do you calculate the Chi-square values for each genotype in a segregating population?

Response: Thanks for your careful review. We calculated the Chi-square values for each genotype using the formula: $\chi^2 = \sum_1^n \frac{(\text{Observe} - \text{Expect})^2}{\text{Expect}}$ based on the previous study

(Yamagata et al., 2010). The χ^2 values of all nine genotypes and the *df* (degree of freedom) were given (please see the revised **Supplementary Table 1**), which refers to the *S27/28* (previously reported two loci contributing to sterility). In fact, the expected plant number was estimated according to the total number of rice plants (1128) in a segregating population, and the sum of the expected number was equal to the sum of the observed number. The numbers 7, 1, and 8 in the column of observed number in **Supplementary Table 1**, which represent the segregation ratio in a selfing progeny of hybrid NIL-*HWS1/HWS2* (Genotype: W1G1|W2G2), disrupted the equality of the sum of the observed number. To make it easier to understand, we have made some appropriate modifications in **Supplementary Table 1**.

Reference:

Yamagata, Y., Yamamoto, E., Aya, K., Win, K.T., Doi, K., Ito, T., Kanamori, H., Wu, J., Matsumoto, T., Matsuoka, M., et al. (2010). Mitochondrial gene in the nuclear genome induces reproductive

barrier in rice. Proceedings of the National Academy of Sciences 107, 1494-1499.

2. In Supplementary Table 2, there are many accessions with more than one copy of RIS1 or RIS2, but the numbers of the Ks value do not completely match the gene copy numbers, such as W1559, W1666, W2051, ...

Response: Thank you for pointing this out. We recalculated the Ks values of *HWS1/2* relative to syntenic orthologs in *O. meridionalis* (an ancient wild rice species with the AA genome) other than orthologs of the original *Leersia perrieri* considering the relatively high level of Ks values. All the copies of *HWS1* and *HWS2* were contained in the Ks calculation (please see the new **Supplementary Table 2**).

3. In Line 150, the word “introgressed” gave me the impression that the null allele of 310 existed in another variety and was crossed into the WYJ7 background, while it was directly created by gene editing instead.

Response: We are sorry for the inappropriate word in use. This misleading word has been corrected to “generated” to avoid ambiguity based on your suggestion (please see **Line 154**); thanks for your reminder.

4. In Line 493, the website “www.RiceRC.com” can not be opened.

Response: Thank you for pointing this out. We have corrected this error and provided the right website address in the revised manuscript (please see **Lines 563-564**).

5. In Line 668-681, the authors didn't specify the materials used in protein extraction and western blotting.

Response: We provided the detailed material information in the method section; thanks for pointing this out (please see **Lines 742-744**).

6. In Extended Data Fig. 1a, “RIS1^{WYJ7}” should be “RIS1^{CG14}”.

Response: We apologize for our careless spelling. We have revised the typo, thank you (please see the revised **Extended Data Fig. 1a**).

7. In Extended Data Fig. 1c, the pollen of CG14 seems to be abnormal. Replace it if you have additional figures.

Response: Thank you for this suggestion. We have taken a new photograph and replaced it (please see the revised **Extended Data Fig 1c**).

8. In Extended Data Fig. 3b, is the tree built based on the EAF6 domain or the whole protein sequence?

Response: Thank you for your question. The tree in **Extended Data Fig. 3b** was built based on the EAF6 domain sequence.

9. In Extended Data Fig. 4a-b, we can determine that RIS genes are constantly expressed in different materials, but it is hard to determine the actual expression level compared to Ubiquitin, because all the relative expressions were normalized twice. More information could be obtained if the authors only normalize the expression data once.

Response: Thank you for your suggestion. Based on your advice, we have reanalyzed the expression data and normalized the gene expression level only once using *Ubiquitin* as reference (please see the revised **Extended Data Fig. 4a, b**).

10. In Extended Data Fig. 6h, there are two “Membrane-bounded organelle” items.

Response: Thank you for your careful reading of our manuscript. We have corrected this GO item (please see the revised **Extended Data Fig. 7h**).

Once again, we would like to take this opportunity to express our gratitude to all of the Reviewers for their careful review and constructive suggestions in improving our work.

REVIEWERS' COMMENTS

Reviewer #1 (Remarks to the Author):

Given the response of the authors, I feel they have satisfactorily addressed my concerns.

Reviewer #2 (Remarks to the Author):

In this revision, the authors well answered the questions and concerns raised by the reviewers. The manuscript was improved largely, including added new experimental results.

Minor revisions:

- 1) The title may be changed to: Dysfunction of a duplicated gene pair for histone acetyltransferases causes segregation distortion and interspecific reproductive barrier in rice.
- 2) Lines 34-35, should be changed to: due to the recombinant HWS1/2-mediated misregulation of

Reviewer #3 (Remarks to the Author):

The authors have added experimental data on the functional validation of the important SNP in HWS1 through promoter activity analysis, and have conducted additional tests on the ILS and introgression hypotheses. I think the HWS1/2 evolutionary history model currently proposed is much more reasonable. For my part, the authors appear to have satisfactorily addressed most of my concerns and revised the manuscript accordingly.

Two minor suggestions:

1. As the authors mentioned in the response letter, " $\chi^2(1:2:1)$ " values are calculated from $(\text{Observe-Expect})^2/\text{Expect}$ of each genotype in Supplementary Table 1. However, the sum of all $(\text{Observe-Expect})^2/\text{Expect}$ values is the real and only one Chi-square value which should be used in significance test. I am still not sure how the "***" significance was calculated for a specific genotype in Fig. 1a and Supplementary Table 1.
2. In lines 377-379, I am not quite sure if the authors suggest that domestication of japonica rice is influenced by bottleneck effect or founder effect or both.

Point-by-point Response to Reviewers

Reviewer #1 (Remarks to the Author):

Given the response of the authors, I feel they have satisfactorily addressed my concerns.

Response: Thanks for your review of our study, and we highly appreciate your positive comments.

Reviewer #2 (Remarks to the Author):

In this revision, the authors well answered the questions and concerns raised by the reviewers. The manuscript was improved largely, including added new experimental results.

Minor revisions:

1) The title may be changed to: Dysfunction of a duplicated gene pair for histone acetyltransferases causes segregation distortion and interspecific reproductive barrier in rice.

Response: Thanks for your suggestion. We have changed the title to ‘Dysfunction of duplicated pair rice histone acetyltransferases causes segregation distortion and an interspecific reproductive barrier’ as recommended.

2) Lines 34-35, should be changed to: due to the recombinant HWS1/2-mediated misregulation of

Response: Thanks for your thoughtful advice. We have modified this in the revised manuscript accordingly (please see **Line 35**).

Reviewer #3 (Remarks to the Author):

The authors have added experimental data on the functional validation of the important SNP in HWS1 through promoter activity analysis, and have conducted additional tests on the ILS and introgression hypotheses. I think the HWS1/2 evolutionary history model currently proposed is much more reasonable. For my part, the authors appear to

have satisfactorily addressed most of my concerns and revised the manuscript accordingly.

Two minor suggestions:

1. As the authors mentioned in the response letter, “ $\chi^2(1:2:1)$ ” values are calculated from $(\text{Observe-Expect})^2/\text{Expect}$ of each genotype in Supplementary Table 1. However, the sum of all $(\text{Observe-Expect})^2/\text{Expect}$ values is the real and only one Chi-square value which should be used in significance test. I am still not sure how the “***” significance was calculated for a specific genotype in Fig. 1a and Supplementary Table 1.

Response: Thank you for pointing this out. Based on your suggestion, we re-calculated the Chi-square value, performed the significance test again, showed the obtained p values and labelled the statistical significance “***” on 81.864 (calculated χ^2 value of the test) (please see the revised **Fig. 1a** and **Supplementary Table 1**).

2. In lines 377-379, I am not quite sure if the authors suggest that domestication of japonica rice is influenced by bottleneck effect or founder effect or both.

Response: Thanks for your comments. We tend to believe that the relatively low nucleotide diversity (π) value for the entire 200-kb region upstream and downstream of *HWS1* locus is attributed to the presence of a bottleneck effect in this region (please see **Fig. 4h; Line 377**). And the high proportion of sterile allele (*hws1^C*) with expression failure in the *japonica* population might be due to the founder effect (please see **Supplementary Table 2; Lines 378-379**, and **Lines 453-454**). Therefore, we incline to the view that the influence of bottleneck effect and founder effect may both exist in the domesticated *japonica* carrying the *hws1^C* allele.

Once again, we would like to express our gratitude to all of the three Reviewers for their continued efforts and insightful suggestions in improving our work. We hope that the second revised manuscript is more acceptable and satisfactory.